# Participatory Arts as Emergency Responses for Strengthening Community Resilience and Psychosocial Support: A Retrospective Phenomenological Inquiry

**DOI:** 10.3390/children12111498

**Published:** 2025-11-04

**Authors:** Konstantinos Mastrothanasis, Cristina Dumitru, Nadina Darie, Maria Kladaki, Emmanouil Pikoulis, Avra Sidiropoulou, Eleni Papouli, Despoina Papantoniou, Anastasia Pikouli, Evika Karamagioli

**Affiliations:** 1School of Medicine, National and Kapodistrian University of Athens, Mikras Asias 75, Goudi, 15772 Athens, Greece; mpikoul@med.uoa.gr (E.P.); evikakara@med.uoa.gr (E.K.); 2Faculty of Humanities & Social Sciences, Open University of Cyprus, 56 Yiannou Kranidioti Avenue, Latsia, 2220 Nicosia, Cyprus; avra.sidiropoulou@ouc.ac.cy; 3Department of Education, Pitești University Centre, National University of Science and Technology Politehnica Bucharest, Târgul din Vale 1, 110040 Pitești, Romania; cristina.dumitru@upit.ro; 4Department of Psychology, Communication Sciences and Social Work, Pitești University Centre, National University for Science and Technology Politehnica Bucuresti, Târgul din Vale 1, 110040 Pitești, Romania; 5Department of Primary Education, University of the Aegean, Dimokratias 1, 85132 Rhodes, Greece; mkladaki@aegean.gr; 6Department of Social Work, University of West Attica, 12244 Egaleo, Greece; epapouli@uniwa.gr; 7Medical School, University of Nicosia, 93 Agiou Nikolaou Street, Engomi, 2408 Nicosia, Cyprus; papantoniou.d1@live.unic.ac.cy

**Keywords:** drama pedagogy, participatory arts, psychosocial resilience, school-based intervention, pediatric well-being, community health

## Abstract

**Background/Objectives:** Public health emergencies disrupt school routines and child development, elevating psychosocial risk. The long-term influence of school-based participatory arts, particularly drama pedagogy, has not been sufficiently explored. This study examined teachers’ retrospective perceptions of the four-year effects of a large-scale, remotely delivered drama-based intervention on children’s psychosocial well-being and school community resilience. **Methods:** We conducted a retrospective interpretative phenomenological study with 23 primary-school teachers who implemented a seven-week, drama-based program with children aged 10–12 during a public health emergency. Semi-structured interviews were conducted four years post-implementation and analyzed following the principles of Interpretative Phenomenological Analysis, using the Community Resilience Framework as a sensitizing theoretical lens. **Results:** According to teachers’ retrospective accounts, participatory arts were perceived to function as a complementary public-health-oriented practice, helping maintain children’s connection to school, and were associated with strengthening trust, creativity, and solidarity, as well as supporting communication, emotional expression, adaptability, and collaborative skills. Teachers reported that stable rituals and drama-based practices appeared to foster a sense of safety amid disruption; over time, some of these practices were reported as becoming part of everyday school routines, which teachers associated with continuity and collective resilience. **Conclusions:** Integrating drama-based interventions into school health and psychosocial crisis-readiness may strengthen pediatric public health strategies and may help education systems to respond to future emergencies. These findings reflect teachers’ perceptions of sustained influence and suggest the perceived value of arts-based methods in developmental/behavioral support and school community resilience. By addressing emotional regulation, peer connection, and psychosocial adaptation within school settings, the intervention reflects the preventive and promotive dimensions of pediatric public health, emphasizing the school’s role as an environment that supports children’s overall mental and developmental health.

## 1. Introduction

Recent large-scale health crises, such as the COVID-19 pandemic, introduced a health and social context that deeply affected school life, interrupting established learning processes and altering children’s everyday routines. The suspension of face-to-face instruction and the forced shift to remote teaching, implemented as emergency distance education, intensified social isolation, increased uncertainty, and aggravated both pre-existing and newly emerging psychosocial difficulties among students [1]. Although initial discussions concentrated on learning gaps and the management of technological transition, it gradually became evident that the more lasting consequences of the pandemic were linked above all to the emotional world of children and adolescents, their social relations, and their capacity for resilience [2].

In this setting, education assumed a role that extended beyond the transmission of knowledge. It became a means of supporting psychosocial well-being and maintaining the cohesion of the school community [3]. In times of crisis, schools become centers of care and creative engagement, where teachers play a key role in strengthening students’ social–emotional learning and resilience [4]. The experience of online learning demonstrated the need for approaches that merge creativity with collaboration and mental health promotion. From a pediatric perspective, creative approaches within schools that combine health promotion with social–emotional learning are closely linked to children’s developmental and behavioral outcomes. This disruption underscored that safeguarding children’s psychosocial and developmental health must be a core component of school health strategies during public health emergencies.

Drama pedagogy, as an applied educational method, may offer a medium for experiential processing, the development of social skills, the development of empathy, and the strengthening of group identity through collective creation and dramatic art [5]. Amid the conditions of the pandemic, the use of drama pedagogy alongside digital technologies provided a framework for teachers and students to engage in creative practices, reflect on outcomes, and manage unexpected difficulties [6,7,8]. The need to maintain interaction, emotional connection, and shared meaning prompted a rethinking of established practices, allowing schools to maintain an active educational environment even under conditions of physical isolation.

While the psychosocial aspects of education during crises have attracted increasing scholarly attention, studies that examine the lasting impact of arts-based and participatory interventions remain relatively scarce [9,10]. Only a limited number of studies examine how such interventions shape, maintain, or transform school life, interpersonal relationships, and children’s adaptive capacities over time [11].

This study seeks to document both individual and collective benefits, to identify the key factors that supported or hindered children’s involvement, and to evaluate the contribution of drama pedagogy to resilience, communicative competence, and psychosocial adjustment under conditions of uncertainty and crisis. The study investigates the long-term effects of participation in an intervention, as recalled and assessed by teachers four years later.

## 2. Education and Arts for Building Resilience in Children

In periods of pandemic or widespread crisis, education changes both as an institutional framework and as a social experience [12]. The Spanish flu of 1918 resulted in school closures and interruptions to teaching. Polio epidemics in the 1950s also led to school suspensions and created concern among communities. Avian flu (H5N1) and swine flu (H1N1/2009) placed schools under public health surveillance and required emergency measures. The 2003 SARS outbreak caused temporary closures in parts of Asia, while Ebola and Zika outbreaks in Africa and Latin America led to long-term interruptions in attendance and increased educational inequalities [13].

The COVID-19 pandemic brought the implementation of universal distance learning, disrupted school routines, and shifted community priorities [14]. During this period, schools assumed additional functions. Beyond their role in knowledge transmission, they also acted as settings of support, points of reference for collective identity, and sites where children’s psychosocial needs were addressed. Studies indicate that students experienced loneliness, disorganization, and loss of routine, together with new forms of digital learning [1,15,16,17,18]. Decisions concerning the reopening or closure of schools, the application of hygiene rules, and adjustments to curricula highlighted long-standing issues of inequality and mental health [2,3,5,9,11,14,16,17,18,19,20,21,22,23,24,25,26,27,28,29,30,31,32,33,34,35,36,37,38,39,40,41,42,43,44,45,46]. Educational practice therefore had to reconsider aims, methods, and structures in order to maintain learning, ensure safety, and provide support under conditions of uncertainty. Within this context, attention to the resilience of school communities became more pronounced.

Resilience, as described in recent literature, does not refer only to the management of a temporary difficulty. It is understood as the capacity of individuals and groups to reorganize, to process adverse experiences, and to develop new strategies for adaptation and problem-solving [41,47]. The *Community Resilience Framework for Disaster and Emergency Preparedness* developed by Pfefferbaum, et al. [48] presents a structured approach to crises and disasters through seven dimensions (see Figure 1).

The first relates to a sense of belonging and the preservation of social ties, even without physical presence. The second refers to the use of available material, institutional, and psychosocial resources that support school functioning. The third concerns the capacity to transform adverse experiences into opportunities for learning, reflection, and shared progress. The fourth emphasizes communication, defined as the flow of reliable information, dialogue, and the management of emotions within the school context. The fifth addresses trust, reciprocity, and cohesion, often described as social capital [49,50]. The sixth concerns flexibility and self-regulation in the face of unpredictability, that is, adaptive skills within the school community. The seventh dimension refers to shared narratives, symbols, and cultural practices that shape interpretations of crises and support collective identity. Taken together, these dimensions present resilience as a process that develops over time, grounded in relationships, collective action, and shared meaning within the school environment.

International experience shows that each period of crisis tends to widen existing inequalities but also encourages the development of new forms of collective practice and innovation in schools, with effects on resilience. Radio education in Chicago in 1937, introduced in response to the polio epidemic, was an early example of alternative instruction through technology [51]. Similarly, during the H1N1 pandemic in 2009, education systems in countries such as Mexico, the United States, and Japan adopted hybrid models of learning and strengthened support networks to reduce isolation [20]. The 2003 SARS outbreak led schools in Hong Kong and Singapore to use artistic and creative activities as methods for addressing collective trauma [52]. Across these cases, the importance of maintaining a school framework that provides stability, support, and opportunities for processing crisis experiences becomes evident.

Drama pedagogy and participatory arts have been described as approaches that may support resilience and addressing psychosocial difficulties in school-aged children, particularly under conditions of uncertainty and crisis [36,39,40]. Recent studies indicate that experiential artistic methods (Readers’ Theatre, dramatic techniques, collective storytelling), can support children’s social, emotional, and behavioral development, create a sense of safety and belonging, and encourage open expression [5,26,35,53,54,55]. However, most of these interventions have been implemented in short-term or context-specific formats, often focusing on immediate psychosocial benefits rather than sustained developmental change. Participation in drama-based interventions may give children the chance to reflect on experiences, address fear and anxiety, cooperate with peers, communicate, and develop creativity and empathy [37,38,42,56]. In this sense, participatory arts complement the principles of frameworks such as CASEL (Collaborative for Academic, Social, and Emotional Learning) and the UNICEF/WHO protocols on Psychosocial First Aid (PFA), both of which emphasize social–emotional learning and the support of school communities in times of crisis [27,45,57,58]. The main goals of such artistic interventions are the strengthening of social, emotional, and academic competences and the management of classroom behavior and peer relations [28]. Factors such as active student participation, cooperation with teachers, and the creation of a stable environment are identified as important for their success. Drama pedagogy appears to contribute to emotion management, the development of social skills, and the strengthening of collective life in schools [52].

The literature on the effectiveness of these approaches in crises during pandemics focuses mainly on short-term outcomes. It highlights improvements in resilience, self-regulation, social engagement, reduction of anxiety and depressive symptoms, as well as increases in self-esteem and school engagement [9]. Research by Giotaki & Lenakakis [54] showed that theatre can assist in dealing with complex social issues and support understanding and empathy, even in young children. Hatton [5] pointed to the role of school drama in fostering critical thinking and shared responsibility in relation to contemporary issues, while Turner-King & Smith [55] demonstrated that applied theatre can provide a framework for addressing crises through context-specific creative activities.

In relation to the COVID-19 pandemic, several recent studies documented the benefits of short interventions for psychological support and school reconnection of children and teachers. Tam’s [9] study in Hong Kong found that incorporating play, drama, and the arts helped rebuild resilience during the transition back to in-person schooling. Cziboly & Bethlenfalvy [7] and Gallagher et al. [8] described how online drama initiatives supported communication, solidarity, and the personal processing of crisis experiences even in distance settings. De Kock et al. [59] emphasized the role of participatory arts in care and community cohesion, while Trezise et al. [60] reported new forms of digital arts education that improved creativity and access.

At the same time, the long-term effects of drama pedagogy remain largely unexplored. The systematic review and meta-analysis by Jiang et al. [9] shows that interventions for mental health and well-being during and immediately after the COVID-19 pandemic were usually short-term (typically 8–12 weeks). Most positive outcomes, such as gains in resilience, emotional regulation, social interaction, and reduced stress, were measured immediately after program ends. The authors underline that long-term effects are still unknown, as follow-up data six months or more after interventions are rare. This lack of evidence highlights the need for systematic investigation into the sustainability of drama-based interventions over time, especially when schools must respond to extended crises. Taken together, these findings illustrate that while drama-based and participatory arts interventions consistently demonstrate short-term psychosocial benefits, their capacity to generate durable behavioral or developmental change remains insufficiently evidenced. The present study therefore responds to this empirical gap by examining teachers’ long-term perceptions of how such interventions may contribute to sustained resilience within the school community.

Despite the numerous interventions introduced during the pandemic, little is known about their long-term consequences, particularly when they integrate health and psychosocial aims through participatory arts [34]. This lack of longitudinal qualitative evidence limits the capacity of educational and health agencies to design sustainable interventions. Systematic empirical research examining the sustainability and long-term contribution of drama pedagogy and participatory arts remains rare. This study addresses that gap by presenting an assessment of the long-term effects of health and psychosocial interventions that use drama pedagogy and participatory arts, with the aim of supporting resilience and psychosocial adaptation of children under conditions of prolonged crisis. By examining experiences four years after implementation, this study provides rare longitudinal qualitative evidence on the sustained influence of participatory arts in crisis-affected schooling.

### Purpose and Research Questions

The purpose of this study is to explore how teachers retrospectively perceived the long-term effects of an emergency remote intervention with health and psychosocial elements, implemented with children aged 10–12 in Greece during a period of prolonged school closures associated with a major public health crisis. The intervention made use of participatory arts, with a particular focus on drama pedagogy. The study seeks to interpret teachers’ accounts of how the intervention was experienced as supporting the development of psychosocial resilience and interpersonal adjustment under conditions of social isolation and uncertainty. It also considers its potential contribution to strengthening pediatric well-being and the ways in which its effects were sustained within school settings over time.

The investigation is conducted four years after the initial implementation of the intervention and draws on the narratives and experiences of the teachers who delivered the program. The aim is to assess the durability of remote response practices and the extent to which they have become integrated into present-day school life. In addition, the study seeks to formulate proposals on how participatory arts and drama pedagogy can be used as tools for educational and psychosocial management in future crises.

On this basis, the study addresses the following research questions:How do teachers describe the intervention’s influence on children’s connection and commitment to their school group four years after its implementation?Which factors or processes as recalled by teachers, appear to continue supporting children’s participation and engagement in school life?In what ways do teachers perceive that involvement in the drama pedagogy intervention shaped children’s approaches to processing and responding to crisis experiences over time?How do teachers interpret changes in children’s communication skills and their ability to express themselves in the longer term?Which elements of the intervention are perceived as contributing to the formation or strengthening of trust and cooperation among children, as described by teachers today?How do teachers describe children’s adaptive capacities following their participation in the program?How do teachers perceive that the arts- and drama-based activities carried out in the intervention influenced the ways children give meaning to and interpret crisis experiences?

This retrospective phenomenological inquiry explores how teachers, four years after the initial implementation, interpret the enduring impact and meaning of the intervention over time.

## 3. Materials and Methods

### 3.1. Research Design

This study adopts a retrospective interpretative phenomenological approach, aiming to explore how teachers, four years later, make sense of and evaluate their experiences from implementing a large-scale emergency remote drama pedagogy intervention introduced during a period of extensive school closures linked to a major public health crisis. The choice of the interpretative rather than the descriptive version of phenomenology was made because the focus of the research goes beyond the detailed recording of experiences [61]. It seeks instead to understand and interpret the meanings that participants attribute to those experiences within the context of their individual and collective narratives. This methodological orientation emphasizes both personal and shared accounts, as well as the interpretation and evaluation of the program over time. The analysis concentrates on teachers’ actual experiences and on how they interpret the changes they observed in children’s psychosocial resilience, interpersonal adjustment, and overall school well-being.

The four-year distance was chosen to allow teachers to reflect on what seemed to remain in everyday school life rather than on short-term impressions. Children were not re-contacted for ethical and practical reasons, as this phase focused on teachers’ retrospective insights.

### 3.2. Brief Description of the School-Based Participatory Arts Intervention for Children

The intervention was grounded in the principles of Social and Emotional Learning (SEL) and resilience, with a focus on supporting children’s psychosocial development and well-being in the school setting [25]. It promoted active participation through drama pedagogy, including Readers’ Theatre based on classic stories and fairy tales, as well as participatory arts and drama techniques (e.g., theatre games, improvisation, process drama) to foster key developmental and psychosocial skills.

The design consisted of a five-day cycle of digital sessions, 20–30 min each day, where activities integrating health-related and psychosocial elements through the arts were delivered in sequence. The sessions aimed to strengthen group cohesion, emotional expression, metacognitive reflection, and collective storytelling. The program was aligned with international standards, in particular the IASC Guidelines on Mental Health and Psychosocial Support in Emergency Settings [23] and the INEE Minimum Standards for Education: Preparedness, Response, Recovery (INEE) [22]. The overall duration of the intervention was seven weeks, totaling 700 min. It was implemented across dozens of schools in Greece [33].

All teachers who took part in the implementation of the intervention had completed a structured training program and received ongoing support from the interdisciplinary scientific team that designed the program. Training and support were provided under the framework of the Centre for Training and Lifelong Learning (KEDIVIM) of the University of the Aegean. The scientific team supplied a detailed implementation guide, structured teaching materials, and systematic monitoring, ensuring consistency, uniformity, and scientific validity in the program’s application across schools.

Health-related activities were defined, in the broad sense of mental health promotion and well-being, as those contributing directly or indirectly to the prevention or management of mental health difficulties, the reduction of dysfunctional behaviors, and the strengthening of resilience under stress (examples: “Group Emotional Pulse,” “Chair of Revelations”). Psychosocial activities were defined as those aiming to enhance social skills, cooperation, solidarity, group cohesion, and social inclusion, without necessarily being linked to mental health, but contributing overall to children’s social and emotional functioning (examples: “Opening Theatre Game,” “Choral Reading,” “Group Monologues,” “Readers’ Theatre”). Many of the activities carried out combined both aspects, contributing simultaneously to mental health promotion and psychosocial support, in line with international evidence-based program requirements. A detailed outline of the weekly sequence of drama-based health and psychosocial activities is presented in Appendix A. The structure illustrates how the program progressively integrated emotional regulation, social learning, and creative expression through drama pedagogy, aligning with international frameworks for mental health and psychosocial support in educational settings.

### 3.3. Participants

In this phase of the study, a total of 23 primary school teachers participated. All had undertaken and successfully completed the remote implementation of the drama pedagogy program for psychosocial support during the COVID-19 restrictions, working with pupils aged 10–12. Of these, 17 were women and 6 were men, teaching in the fourth, fifth, and sixth grades of primary school, as well as in inclusion classes or priority education zones (ZEP) during the period of the intervention. Regarding their educational background, nearly 65% of participants held a postgraduate degree. Most were permanent teachers, with an average teaching experience of about nine years. In total, 239 teachers were invited to participate.

The selection of participants followed a purposive sampling strategy. The aim was to include teachers with documented experience of delivering this complex intervention during the study period, who had also maintained systematic contact with the children involved during the restrictions and in subsequent years. Initial communication with potential participants was carried out by email, using the contact information recorded during the intervention phase and only where prior consent for future communication had been provided. The final sample included all those who responded positively to the invitation. The participants represented schools from both urban and rural areas across five educational regions, ensuring diversity in socioeconomic and institutional contexts.

### 3.4. Instrument

For the collection of qualitative data in this study, a semi-structured interview was used. It was designed specifically to investigate the long-term effects of the drama pedagogy intervention, four years after its initial implementation. In total, the interview guide consisted of thirty questions. These were developed to encourage detailed accounts of the experiences, attitudes, and evaluations of the participating teachers regarding different aspects of the intervention and the resilience of the school community under crisis conditions (see Appendix B).

The structure and content of the interview guide were based on the theoretical framework of Pfefferbaum et al. [48]. The guide was organized into seven thematic areas, corresponding to the framework’s main dimensions: connection and commitment, support resources and facilitating factors, transformative capacity, information and communication, social capital, adaptive capacity, and cultural or symbolic meaning-making.

### 3.5. Procedure

For the collection of qualitative data in this study, a semi-structured interview was used. Approval for this phase of the study was obtained from the Ethics and Research Committee of the Medical School of the National and Kapodistrian University of Athens (Ref. No. 242/18 June 2025). Each participant received a detailed information sheet outlining the purpose, methodology, and voluntary nature of the study, with particular emphasis on confidentiality, anonymity, and the right to withdraw at any time without consequence. Participation proceeded only after written informed consent was obtained.

The interviews were conducted online at a time agreed with each teacher, lasted approximately 45–60 min, and were audio-recorded with participants’ consent. All recordings were fully anonymized before transcription and analysis. To ensure anonymity, identifiable information (such as names, schools, or specific local references) was removed or replaced with coded identifiers during transcription, and the key linking codes to participants was permanently deleted after verification.

### 3.6. Analyses

The recorded data were transcribed verbatim, anonymized, and entered into QCAmap software (version 1.2.0) [62], which supported data organization and retrieval. The analysis followed the principles of Interpretative Phenomenological Analysis (IPA) [63] aiming to explore how teachers made sense of their lived experience of implementing the drama-based intervention and the meanings they attribute to its outcomes.

The process combined descriptive, linguistic, and conceptual levels of interpretation. Each transcript was first read multiple times to achieve immersion, followed by initial noting and the identification of emergent themes closely tied to the participant’s words. Each case was analyzed idiographically before cross-case synthesis, preserving the individuality of each teacher’s experiential account prior to identifying shared interpretative patterns. These themes were then clustered into superordinate themes to reflect shared patterns across participants while maintaining idiographic sensitivity.

The Community Resilience Framework for Disaster and Emergency Preparedness [48] served as a sensitizing theoretical lens, guiding interpretative clustering without functioning as a predetermined codebook. Although the same dataset was re-examined, the interpretative focus shifted from descriptive theme classification to the exploration of meaning construction through an iterative hermeneutic cycle, consistent with IPA principles. Data collection and analysis proceeded iteratively until thematic saturation was reached, as no substantially new interpretative themes emerged in the final interviews, indicating stability and completeness in the analytic categories.

The interpretative coding was carried out independently by two researchers. Cohen’s kappa (value = 0.76) indicated satisfactory agreement in the identification and grouping of meaning units. In cases of divergence, a reflexive dialogue process was used until consensus was reached regarding the final organization of interpretative themes. Attention was also given to divergent or disconfirming accounts, which were comparatively examined to ensure that the final interpretative structure reflected the full range of teachers’ perspectives and meaning variations.

## 4. Results

The IPA process generated a total of 59 meaning units, which were progressively organized into 31 emergent themes and ultimately synthesized into seven superordinate interpretative categories, corresponding to the dimensions of the theoretical framework. The presentation that follows is structured according to these interpretative categories, with systematic documentation through illustrative meaning units and interpretative descriptions, highlighting shared patterns and experiential nuances in teachers’ accounts.

Teachers’ accounts include both recollections of specific moments during the original intervention and reflections on how these experiences have continued to influence students and school practices four years later. The distinction between recalled experiences and perceived sustained effects is noted where relevant.

### 4.1. Connection and Commitment Within the School Group

The first superordinate interpretative category concerns the sense of connection that participants attributed to individuals’ relationships with the social, school, and wider community context, as retrospectively interpreted four years after the remote intervention (see Table 1). One of the main meaning patterns articulated by participants was the recollection of a sense of group identity formed through the regular and repeated rituals of the intervention (e.g., structured openings with theatre games, collective storytelling). Teachers recalled these experiences as characteristic of the initial implementation period, while they also perceived that some of the resulting group practices and rituals persisted in school routines years later. This was recognized as an important factor in maintaining children’s link with the school environment during the pandemic (“*It was one of the few times that children spoke about school with such enthusiasm, even though they were physically distant*” Participant 5).

The development of an alternative form of school identity, which arose from children’s consistent participation in digital activities, was also reported as contributing to the preservation of group cohesion. Teachers perceived that certain elements of the practice appeared to have become part of everyday school routines, such as weekly drama sessions or symbolic openings that continued to be used after the program (“*Even now, after such a long time, they still remember the Monday ‘theatre morning’… one of my students mentioned it recently*” Participant 13).

The children’s commitment to the group was expressed through consistency and anticipation for the activities, despite the practical difficulties of the remote setting (“*They did not want to miss a single session, even if at times their cameras were off. They felt part of a group…*,” Participant 7). During the sessions, children often assumed specific roles, which strengthened both their individual presence and the collective interaction.

A further element highlighted was the communal dimension of gathering through the screen, with virtual presentations and the participation of parents acting as digital points of reconnection with the school and the wider community. Finally, the retrospective sense of security provided by the stable structure of the intervention was noted as an important element of psychosocial support (“*This routine, knowing that every day followed the same pattern, gave children a sense of reassurance*” Participant 11). This contributed to resilience and supported children’s adjustment during a period marked by instability and uncertainty.

### 4.2. Support Factors and Participation in School Life

The superordinate interpretative category relates to the factors and processes that shaped and continue to support children’s participation and psychosocial well-being in school life, as recalled and evaluated four years after the intervention (see Table 2). Several accounts emphasized the importance of access to and support with digital tools, both at the school and family level, as a precondition for maintaining participation during isolation (“*Having relatively easy access to the platform and parental help made a real difference*” Participant 2).

Equally significant was the structured material and detailed guide provided by the program, which covered both psychosocial and health-related activities. This resource reduced teachers’ stress and ensured consistency in implementation (“*The material guided us step by step. It was a detailed teacher’s manual. I don’t recall ever being left without support*” Participant 14).

Ongoing guidance and the continuous presence of the interdisciplinary team, together with mutual support among participating teachers, strengthened teachers’ sense of security and readiness, which in turn had a stabilizing effect on students. The presence of a stable routine, with repeated structures and activity patterns, also created a sense of psychological safety and continuity, facilitating children’s adjustment and engagement (“*Repeating the same program daily gave reassurance to both children and us*” Participant 9).

Encouragement, regular acknowledgment of effort, and positive feedback were also identified as important for sustaining interest and participation. At an institutional level, continued training and the creation of informal teacher networks supported the exchange of good practices. Some practices were retained by teachers and integrated into their day-to-day teaching, facilitating the gradual incorporation of the intervention’s ethos into school life (“*Most of us kept using parts of the program in later years. For example, I often apply Readers’ Theatre in teaching history*” Participant 8).

Finally, the collective nature of artistic activities played a key role in building lasting bonds, reinforcing children’s sense of belonging, and sustaining their interest in joint work. Regular participation and the assumption of small roles, often in the context of Readers’ Theatre, further encouraged personal involvement and long-term engagement with school life (“*They waited for their turn, and that gave them motivation to join every time*” Participant 12).

### 4.3. Processing Crisis Experiences

The third superordinate interpretative category focuses on children’s capacity to transform crisis experiences into opportunities for learning and development, as described by teachers four years after the intervention (see Table 3). Participants noted that the drama pedagogy program provided children with a safe and symbolic means to express feelings and concerns. As one teacher explained, “*Taking on roles helped them talk about what they were going through without feeling exposed*” (Participant 4).

Crisis experiences were processed through dramatization, Readers’ Theatre, storytelling, and role adoption, allowing children to reflect on both collective and individual aspects of the pandemic (“*The children often spoke as characters, but in a way they were voicing their own thoughts*” Participant 13). Such recollections refer to moments from the original intervention that teachers still remember as particularly meaningful, which they associate with children’s continued ability to express emotions more openly in later school years. The methods also strengthened children’s confidence in communication and emotional expression. Teachers reported that even the more reserved students began to speak and participate more actively.

Symbolic meaning-making was also central. Drama pedagogy enabled children to reinterpret the pandemic experience by drawing on myth, narrative, and creative activity.

The intervention further supported resilience and coping skills. Through role play and experiential artistic activities, children could work through difficult experiences and test alternative ways of responding in a protected, symbolic setting. The experience of role-taking acted as a safe space to try out emotions, behaviors, and possible solutions, fostering self-awareness, flexibility, and adaptability (“*They learned to look for solutions, endure difficulties, and ask for help when needed*” Participant 9). Teachers perceived that some of these skills seemed to remain visible in children’s current school life (“*Once children acquire them, they tend to continue using them*” Participant 3).

### 4.4. Communication and Expression

The fourth superordinate interpretative category concerns communication and expression, specifically how participation in the drama-based intervention contributed to the development of communicative skills and broadened children’s expressive capacities, as assessed four years after the program (see Table 4). A key point emerging from the narratives is the cultivation of oral language and narrative ability. Teachers noted that pupils developed clearer, more structured speech, alongside greater fluency in recounting experiences and emotions. As one teacher observed, “*Several children began to express themselves more easily, to speak without fear about what they felt*” (Participant 8). The experience of public narration, individually or in groups, provided practice in organizing speech and achieving clarity.

Another recurring theme is the externalization and regulation of emotions. Teachers emphasized that program activities encouraged children to openly express feelings they previously found difficult to communicate: “*We saw many children saying ‘I feel sad’ or ‘I was afraid,’ something they used to struggle with*” (Participant 5). Roles and symbolic stories created a safe framework for handling inner tensions and concerns.

Teachers also highlighted the emergence of meaningful dialogue and interaction among peers, with children engaging more consistently in group discussions and showing improved listening, responding, and turn-taking skills.

Finally, the intervention fostered communicative self-confidence. Overcoming initial hesitation, speaking publicly, and presenting before classmates, parents, or teachers laid a foundation of lasting confidence that, according to participants, still benefits the children today, often reducing their anxiety about public exposure: “*Speaking in front of parents or performing, even through a screen, gave them the courage to keep expressing themselves later on*” (Participant 2). Teachers emphasized that this confidence, first observed during the intervention, appeared to be sustained when they later encountered the same students.

### 4.5. Trust and Cooperation

The fifth superordinate interpretative category focuses on the relationships of trust and cooperation that were formed or strengthened through participation in the drama-based intervention, as retrospectively assessed today (see Table 5). According to participants’ accounts, collaborative work and the pursuit of shared goals within the intervention served as a key mechanism for reinforcing group identity. Activities requiring cooperation, joint preparation, and collective presentation highlighted the importance of mutual support (“*They helped each other remember their roles or lines; this bonded them as a group*” Participant 7).

Encouragement of open communication and sincere sharing of experiences fostered an atmosphere of trust, as children felt free to express personal thoughts and emotions, knowing that the group functioned in a supportive and protective way. At the same time, emphasis was placed on the acceptance of diversity, both in terms of opinions and personal traits, which, as noted, led to a more tolerant and cohesive classroom environment (“*Every child had their own voice, and this was respected…*” Participant 3).

Participants recalled that some interpersonal relationships formed during the intervention appeared to persist over time (“*Even now, in secondary school, they collaborate on projects and support each other*” Participant 13). The experience of solidarity became particularly evident in cases where children supported peers facing difficulties during the period of isolation.

The narratives underscore the collective problem-solving process and the active involvement of all children in overcoming difficulties encountered during the theatrical activities. Strengthening these social bonds and fostering trust and reciprocity were described by participants as one of the most lasting and meaningful outcomes of the intervention (“*The group worked like a safety net. The children do not forget this*” Participant 11).

### 4.6. Adaptation to New Circumstances

The sixth superordinate interpretative category focuses on the adaptive abilities developed by children through their participation in the remote drama-based intervention, particularly in terms of flexibility, self-regulation, and the capacity to handle new or unpredictable situations (see Table 6). Narratives collected four years later highlight that repeated exposure to change, whether program adjustments, technical obstacles, or shifting roles within activities, helped cultivate adaptability and resilience. As one teacher explained, “*They learned to expect that something might change at the last minute, and they handled it better*” (Participant 8).

Technical difficulties, frequent in the online setting, also provided opportunities to practice accepting uncertainty, encouraging calm responses, and fostering both autonomous problem-solving and help-seeking behaviors.

The intervention further encouraged initiative-taking and self-regulation. Children practiced managing emotions in demanding moments and contributed actively to solving both collective and individual challenges. “*They took the initiative to help others or to try something new without fearing mistakes*” one teacher remarked (Participant 12). Teachers interpreted such memories as formative experiences that contributed to adaptive behaviors still observable among the same children four years later.

Psychological flexibility was also strengthened through role changes and exposure to different perspectives, which allowed children to adapt quickly to shifting demands and develop alternative problem-solving strategies. “*Switching roles or finding a different way of doing things taught them to think more openly*” a participant noted (Participant 16). Lastly, the process fostered greater tolerance for uncertainty. Over time, children became accustomed to unexpected changes while maintaining both engagement and functionality.

### 4.7. Meaning-Making and Interpretation of Experiences Through Performing Arts

The seventh superordinate interpretative category focuses on the cultural and symbolic meaning-making of crisis experiences, namely, the ability of children to draw meaning and process the lived reality of the pandemic through cultural and symbolic forms of expression, such as art, theater, and storytelling (see Table 7). Testimonies from participants highlight that theatrical activities functioned as a key medium for transferring both personal and collective experiences into a safe symbolic framework, while simultaneously strengthening hope, transcendence, and collective identity.

Several educators recalled the use of theatrical metaphors and structured narratives, which allowed children to articulate thoughts, fears, questions, and aspirations without direct personal exposure (“*In stories and role-playing games, children found a way to talk about what concerned them, even if not directly*” Participant 23). Fairy tales, myths, and dramatized storytelling were identified as essential tools for processing individual experiences and embedding them into a broader collective narrative.

Particular attention was given to the role of art as a symbolic outlet and a source of hope. Through theatrical actions, children were able to imagine alternative outcomes and express expectations for the future (“*In the plays we staged, there was always a way to arrive at something better, even if we started from difficult themes*” Participant 9). The journey from trauma to collective creation was acknowledged by many teachers as a process that empowered the group and restored lost optimism. At the same time, children’s personal experiences were integrated into the collective narrative of the class, often through theatrical improvisations or identification with characters, allowing a shared form of processing lived experiences. Thus, art functioned not only as a supportive element but also as a central cultural medium of empowerment and crisis management.

Lastly, participants considered that the integration of artistic and drama-based approaches within structured health and psychosocial interventions could serve as a model for future educational policy during pandemics and periods of social isolation. These reflections combined memories of the original performances with later observations of how artistic collaboration continued to inform school culture and collective attitudes. Participants commonly reflected on the perceived power of art to sustain collective identity and resilience within the school context (“*The experience of performance and collaboration through art should be part of every school, especially in difficult times*” Participant 17).

## 5. Discussion

This study examined the long-term effects of a remote drama-based intervention combining psychosocial and health-related artistic activities, which was implemented with students aged 10–12 during the COVID-19 pandemic in Greece. Drawing on the perspective of the Community Resilience Model, the research sought to highlight how participating teachers retrospectively evaluated, four years later, the intervention’s perceived contribution to strengthening children’s psychosocial functioning, social inclusion, and resilience within their school communities. Through qualitative analysis, teachers’ experiences and reflections were recorded, with particular emphasis on the core dimensions of the theoretical framework and the sustainability of the outcomes in educational practice.

### 5.1. The Significance of Connection and Commitment in the Post-COVID School Reality

With regard to the first research question, the findings of this study indicate that the drama-based intervention contributed to the long-term maintenance and reinforcement of children’s connection and commitment to the school and community environment, even under conditions of prolonged isolation during the pandemic. Teachers’ references to the revival of a sense of group belonging (consistent rituals, digital encounters) are in line with previous studies highlighting the role of participatory arts practices in strengthening belonging [5,7,8,54,55]. As Pfefferbaum et al. [48] emphasize, collective identity and stable routines are fundamental elements of community resilience, while the alternative school identity generated through online activities illustrates the transformative role of art and theatre as core components of psychosocial support. These accounts suggest that such outcomes may not be limited to the short term but were remembered by teachers as a lasting “memory code” for students [44]. The reinforcement of collective identity, continuity, and a sense of safety through structured drama-based practices emerges as a central factor for the long-term psychosocial resilience of the school community.

The commitment and consistency shown by students, despite the absence of physical presence, may be interpreted through the lens of social identity theory [24]. Children invested emotionally in the group and maintained active participation through their anticipation of the collective artistic activities. This finding resonates with the results of Mastrothanasis, Pikoulis et al. [36], who underline that collective bonds can be sustained even in digital environments when strong relationships are formed through shared action and participation.

Equally important was the role of digital presentations and virtual community gatherings as “symbolic nodes” of reconnection, confirming previous research that emphasizes the importance of community and family as protective factors in times of crisis [29,32]. The retrospective sense of safety attributed to the stable structure of the intervention is consistent with studies that highlight the necessity of routines as a fundamental support for children’s resilience [48,50].

### 5.2. Sustainability Factors in Student Participation and Support

With regard to the second research question, which focuses on the factors and processes that continue to support children’s participation and psychosocial wellbeing in school life, the findings of this study illustrate a dynamic grounded both in structural and experiential elements of the intervention. First, technical support and guaranteed accessibility emerged as fundamental prerequisites for consistent participation, a point also confirmed in the literature on inequalities in remote education and the significance of technological inclusion [36]. The active support of parents proved important for sustaining engagement, confirming research highlighting the protective function of the family environment during the pandemic [18].

Equally important was the availability of a structured support material (detailed guides, organized activities), which reduced anxiety and enhanced teachers’ confidence by offering a unified framework for implementation. This element represents a basic success factor in innovative interventions, especially in times of crisis [64]. Continuous training, the opportunity for ongoing guidance, and the network of peer support among teachers were also important in maintaining coherence within the intervention, facilitating the dissemination of good practices, and reinforcing professional resilience [48,50,65,66,67].

The institutionalization of consistent routines and consistency in the duration of activities enhanced a framework of safety that encouraged psychological stability and active student involvement. International evidence underscores that environmental stability and repeated routines are central to resilience in conditions of crisis. As highlighted by Selman & Dilworth-Bart [68], the presence of consistent routines is directly linked with improved psychological adjustment and reduced anxiety symptoms among children. Equally decisive was the role of acknowledging effort and providing systematic feedback. Research shows that structured positive feedback enhances intrinsic motivation and students’ self-esteem [69]. Teachers emphasized that appropriately framed feedback not only sustained children’s commitment to the learning process but also strengthened their sense of emotional security and self-confidence, particularly in periods of crisis.

Finally, collaborative participation through group activities and the distribution of small roles fostered an environment of collectivity and belonging that persisted beyond the intervention. These findings, as perceived by teachers, highlight the potential importance of social bonds and active participation for the longer-term strengthening of school life in comparable contexts, while also aligning with contemporary applications of drama pedagogy designed to activate participation and sustain collective identity [10,52].

### 5.3. The Role of the Intervention in Crisis Management and Personal Development

Regarding the third research question, which focused on how engagement in the drama-based intervention supported children in processing and managing experiences of crisis, the findings revealed multilayered processes that strengthened both individual and collective resilience. Participation in dramatic and narrative activities enabled children to express and work through difficult experiences in a safe and symbolic way. Assuming roles functioned as a “testing ground” for emotional regulation, while narrative representation allowed them to reframe experiences, reducing anxiety and uncertainty. These observations are attuned with international studies emphasizing the role of art and drama as protective tools for trauma processing through symbolic expression and role-taking [19,23,30,51,57].

The strengthening of personal voice and meaningful role-taking further contributed to self-confidence and self-expression. Students became more skilled at communicating emotions and thoughts, while also recognizing both the individual and collective dimensions of crisis. These findings align with prior studies indicating that drama pedagogy can promote self-awareness, empathy, and communicative competence in crisis contexts [36,54,55].

Equally important was the development of adaptive skills. Children experimented with alternative strategies for problem-solving, cultivating flexibility and resilience that facilitated adjustment to uncertainty. These findings converge with research linking drama and artistic engagement to improved adaptability and stress management among young people [9,56].

Lastly, myths, stories, and symbolic narratives functioned as interpretative frameworks, enabling children to place crisis experiences within a broader context of collective memory and learning [21,70]. This symbolic meaning-making aligns with perspectives on community resilience, provides children with ways to renegotiate experiences of trauma and transform vulnerability into competence.

While teachers’ recollections were predominantly positive, this pattern may also reflect contextual factors beyond the drama pedagogy itself. The novelty of the remote artistic experience during the crisis, together with the structured guidance and sense of collective support provided to educators, may have contributed to the enduringly favorable perceptions reported. Nevertheless, the teachers’ consistent emphasis on symbolic expression, collaborative creativity, and emotional regulation suggests that drama-based methods proposed a framework for coping and reflection. Recognizing these complementary explanations improves the interpretative depth of the findings and underlines the complex, multi-layered nature of memory and meaning-making in post-crisis school settings.

### 5.4. Long-Term Effects on Communication Skills

The fourth research question examined how participation in the drama-based intervention influenced children’s communication and expressive skills over time. Teachers reported sustained improvement across multiple dimensions. Oral fluency and narrative competence improved, reflected both in teachers’ reports and in students’ growing ability to articulate experiences and emotions clearly. This progress is linked to systematic exposure to storytelling, public speaking, and group presentation practices inherent in drama pedagogy. The use of role-play and narrative mediation provided a safe framework for expressing even ambiguous or challenging emotions [5,57].

The intervention also promoted greater emotional articulation. Students became more willing to discuss personal concerns openly, using theater, drama and symbolic narratives as a medium for processing inner tensions. The symbolic function of role-taking, through identification with fictional characters or dramatized dialogues, proved effective in releasing tension and deepening self-awareness [9,43]. Interpersonal communication also benefited. Students learned to collaborate meaningfully, listen actively, and engage constructively with peers and teachers, which reshaped their approach to group discussions and theatrical presentations while promoting inclusivity and acceptance of diverse perspectives [34].

Lastly, the intervention strengthened communicative confidence. Regular exposure to public performance reduced hesitation and performance anxiety, presenting a safe environment for practicing self-expression. These gains, as noted by teachers, have long-term impact, supporting broader development of confidence, social skills, and self-regulation [71,72].

### 5.5. Formation of Social Capital

The fifth research question examined how participation in the drama-based intervention contributed to building trust and cooperation among children. The findings showed multi-layered processes of social capital formation with long-term effects. Collective goal achievement and group preparation of activities strengthened team identity and cohesion. Ongoing peer support during performances and group work promoted reciprocity and trust, factors linked in the literature to enhanced psychological safety and collaborative skills [8,54,55].

Also noteworthy was the cultivation of open communication and acceptance of diversity within the group, creating a safe environment for children to share thoughts and emotions without fear of criticism. Mutual respect and inclusivity, as highlighted in international research, are important for cohesive and tolerant school communities [30].

Social relationships developed during the program often extended beyond the school setting. The persistence of new bonds after the intervention reveals the durability of outcomes and supports claims that social capital generated through participatory arts has lasting value. Experiences of solidarity and collective problem-solving, both during isolation and in theater activities, strengthened the sense of “shared destiny”. This finding resonates with studies on social capital in pandemics [50,66], underscoring cooperation and mutual support as key elements of school resilience in times of crisis.

### 5.6. Strengthening Adaptability and Psychosocial Flexibility

The sixth research question focused on how children’s ability to adapt to new or difficult situations was formed after participating in the program. The findings indicate that experiential engagement in the drama-based intervention significantly strengthened adaptive skills and psychological flexibility. Teachers noted that children gradually became more familiar with unpredictability.

Self-regulation also emerged as an important factor, with children learning to manage emotions, take initiative, and support peers. This growth was linked to the participatory nature of the activities, where students assumed shifting roles and navigated varied emotional demands [31]. The intervention thus provided a safe environment for experimenting and gradually strengthening self-awareness and internal control.

Psychological flexibility was further enhanced by role changes, exposure to multiple views, and the constant need to adjust to activity requirements [73]. This ability to develop alternative coping strategies represents an important resource for their school and social life. Lastly, the capacity to manage uncertainty was reinforced by cultivating a sense of control and self-efficacy. Children sustained focus and functionality despite shifting conditions, which educators directly associated with improved resilience and adaptability as essential life skills [3,21,74].

### 5.7. Arts and Drama/Theatre as Means of Meaning-Making and Empowerment During Crises

The seventh research question explored the role of arts and theater in helping children interpret and give meaning to crisis experiences. The findings highlight benefits that extend beyond standard psychosocial support. According to the participating teachers, drama-based activities provided children with a safe and creative framework where complex or even traumatic experiences could be translated into symbolic forms of expression. The use of theatrical metaphors and narrative structures, such as myths or fairy tales, enabled emotional processing without the stress of personal exposure, thus facilitating both individual and collective “normalization” of the crisis experience [21].

Another important aspect was the development of hope and a sense of overcoming. Through roles and dramatic action, children were encouraged to imagine alternative outcomes, articulate expectations, and sustain a sense of perspective. This process, as teachers described, helped counter feelings of collective helplessness and fostered optimism. The shift from personal trauma to collective creation effectively functioned as a mechanism of “social processing” of the crisis [10].

Important was the integration of individual experiences into a shared narrative too. Many children were able to embed their personal emotions within group activities of a psychosocial or health-related nature, whether through improvisation or identification with theatrical characters. In this way, the crisis experience was reshaped into a collective story, which strengthens meaning-making and reinforcing the cohesion of the school community [5].

Lastly, the findings underline the broader significance of collective art forms for educational policy and long-term resilience. Teachers underlined that the artistic practices adopted during the intervention can serve as a model for future programs in contexts of health emergencies, as they sustain collective identity while fostering a “cultural memory” that supports community resilience. In line with previous research, this study affirms that the symbolic and narrative tools of art provide children with constructive ways to process crises and reinforce cohesion, solidarity, and hope within school communities [9,75].

### 5.8. Pediatric and School Health Implications

For pediatric public health and school health services, these findings suggest, based on teachers’ retrospective accounts, that structured drama-based routines may be feasibly integrated into everyday practice and may help support developmental/behavioral domains (communication, self-regulation, peer interaction) while potentially buffering psychosocial risk during disruptions. From a pediatric health perspective, these outcomes correspond to core protective factors for children’s mental and emotional well-being (e.g., stability, connectedness, emotional expression) which are recognized within preventive child health frameworks as essential for developmental balance and resilience. Practical implications include embedding brief drama-based activities into school nursing and psychosocial support pathways, using them as universal preventive tools as well as selective supports for vulnerable pupils, and aligning them with SEL frameworks and referral networks. This arts-in-education strategy may strengthen preparedness for future public health emergencies while contributing to equity in access to psychosocial care at school.

### 5.9. Limitations and Directions for Future Research

The study’s main limitation concerns its use of qualitative data provided by teachers, a choice that may have influenced the interpretation of findings, given that the conclusions draw mainly on the implementers’ perspective and not on the experiences of students or others (e.g., parents). However, this approach was deliberately chosen to capture long-term insights from those who directly implemented the program and maintained systematic observation of students’ progress over time. Their ongoing engagement allowed for more reliable retrospective reflection.

Moreover, the sample was restricted to individuals with experience in the intervention and ongoing contact with participating students, which may reduce the generalizability of findings to broader school contexts or different social conditions. Because participation was voluntary, a degree of self-selection bias cannot be excluded; nevertheless, efforts were made to ensure diversity in teaching experience and school type, and data saturation was used as a safeguard for representativeness.

Data collection through retrospective accounts also raises concerns about memory bias and reconstruction of past experiences [76]. To minimize this limitation, interview prompts were anchored in specific events, routines, and activities of the original intervention, helping participants recall concrete details rather than generalized impressions.

The retrospective design also limits the possibility of causal or comparative claims; therefore, findings were interpreted as perceived long-term influences rather than direct effects.

The study did not employ triangulation through multiple data sources (e.g., parents or students), which could have enhanced validity; however, the convergence of responses across teachers and the consistency of recurring themes were used as indicators of analytic credibility.

Although the interpretative phenomenological approach precludes the use of control groups, methodological rigor was maintained through systematic coding by two independent researchers. The degree of agreement between their coding was assessed using Cohen’s kappa, indicating substantial interrater reliability and reducing the likelihood of investigator bias in data interpretation.

Another limitation is that, although the intervention itself provided an indirect form of psychosocial support, there was no parallel structured plan to address potential stress or emotional burden on teachers during data collection. To mitigate this risk, supportive communication was maintained throughout the process, ensuring participants could withdraw or pause at any stage. Future designs could incorporate access to counseling resources or referral mechanisms, ensuring that participants are supported when reflecting on crisis-related experiences.

Future research could broaden the methodology by including multiple data sources (students’ or/and parents’ perspectives) or by applying mixed-method approaches to strengthen validity and scope. It would also be valuable to explore how internationally recognized social–emotional learning programs (e.g., CASEL or SEL frameworks) integrate drama pedagogy and assess their impact on both the sustainability of interventions and the strengthening of students’ psychosocial resilience during crises and pandemics. Additionally, future studies could investigate how participatory arts-based interventions may be adapted within broader pediatric public health frameworks, examining their contribution to emergency preparedness, school health policy, and long-term psychosocial care systems in educational contexts.

Because the present evidence derives from retrospective teacher narratives rather than direct assessments of children, the findings should be interpreted as perceived long-term influences on students’ functioning and school culture rather than as objective measurements of current pediatric outcomes.

## 6. Conclusions

The findings of this study indicate that systematically integrating drama pedagogy was described by teachers as supporting students’ psychosocial well-being while also helping maintain the cohesion of the school community amid uncertainty and social isolation. Analysis of teachers’ narratives revealed that participatory art forms, centered on drama-based approaches, provided what teachers perceived as safe and creative spaces where they could process and make sense of crisis experiences, strengthen resilience, and develop essential adaptive and collaborative skills [8].

The findings highlight, from the teachers’ perspective, the potential value of designing interventions that address both health and psychosocial needs through participatory art practices, particularly drama pedagogy, which were perceived to foster support and collective empowerment [9]. At the level of educational policy, the pandemic illustrated the relevance of integrating creative and participatory approaches into school-based psychosocial frameworks, as perceived by participants. Developing collaborations between schools, public health authorities, and cultural organizations, as suggested by participants, could be further explored as a means to promote resilience and supportive school environments in comparable contexts. These conclusions reflect teachers’ long-term perceptions within a specific national setting and do not establish causal effects or directly measured child outcomes.

Ultimately, this study points to the need for further exploration of the long-term impact of drama pedagogy interventions, both in terms of their sustained effectiveness and the factors that shape their transferability across different educational and cultural settings. In light of continuing uncertainty and the growing demands that emerge during health crises, contemporary pedagogical strategies for rapid response may benefit from recognizing and harnessing the potential of the arts as a means of supporting collective resilience and psychosocial care. Viewed from a pediatric public health standpoint, these findings also suggest that creative, school-based arts initiatives can play a role in promoting children’s overall well-being. By nurturing emotional regulation, adaptive coping, and a sense of shared safety, such approaches contribute to the preventive and developmental dimensions of child health that underpin resilient school communities.

## Figures and Tables

**Figure 1 children-12-01498-f001:**
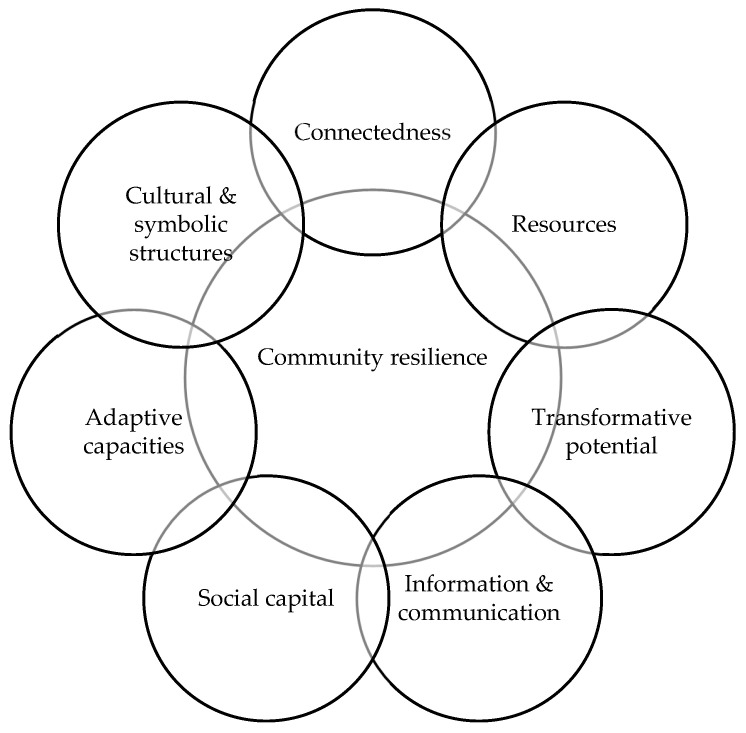
Community resilience framework for disaster and emergency preparedness.

**Table 1 children-12-01498-t001:** Superordinate and emergent meaning reflecting connection and commitment within the school and community context through drama pedagogy interventions.

Emergent Themes	Illustrative Meaning Units	Interpretative Description
Recall of group belonging	Memory of participation in collectivity	Sense of belonging to a group, even in an online environment.
	Stability through rituals	Children recall and establish small ritual practices within a safe framework.
Reclaiming relationship with school	Alternative “school identity”	The intervention was experienced as a distinct school activity, cultivating a different kind of bond with the school group.
Commitment despite distance	Ongoing participation despite distance	Sustained participation and continuity within the group.
Recognition of special roles	Activation of “school memory”	Specific moments that remained in memory.
Community gathering through screen	Digital collectivity	Virtual presentations functioned as moments of reconnection.
Retrospective sense of security	Memory of participation in collectivity	Sense of belonging to a group, even in an online environment.

**Table 2 children-12-01498-t002:** Superordinate and emergent meaning patterns on support factors and participation in school life.

Emergent Themes	Illustrative Meaning Units	Interpretative Description
Technical support	Access to technology	Availability of technological tools for participation
	Support from parents	Parents’ involvement in facilitating participation
Support material and guidance	Detailed guide	Step-by-step instructions for implementation
	Activity material	Ready-to-use activities for facilitation
Training and networking	Ongoing guidance	Continuous support from a team of specialists
	Informal teacher network	Sharing of experiences and good practices
Structure and consistency	Stable routine	Repeated patterns providing security and readiness to respond
	Continuity over time	Consistent application reinforcing a sense of continuity
Encouragement and feedback	Recognition of effort	Emphasis on acknowledgment and positive reinforcement
	Regular feedback	Frequent feedback to strengthen engagement
Collaborative participation	Group activities	Appearance of collectivity and cooperation
	Small roles	Assignment of roles that promoted active participation

**Table 3 children-12-01498-t003:** Superordinate and emergent meaning patterns illustrating the intervention’s transformative potential.

Emergent Themes	Illustrative Meaning Units	Interpretative Description
Processing Difficult experiences	Reflection on crisis	Children expressed and worked through difficult experiences via roles and narrative activities.
	Externalizing emotions	Symbolic representation through theatre forms supported release of emotions and management of stress and uncertainty.
Strengthening personal voice	Encouraging self-expression	Active participation increased self-confidence and freedom in emotional expression.
	Role-taking with meaning	Adopting roles helped children recognize personal and social dimensions of the self.
Symbolic meaning-making	Creating new narratives	Drama pedagogy practices gave new meaning to crisis experiences, turning trauma into learning.
	Using myth and storytelling	Classical stories and myths offered a safe symbolic frame for processing experiences.
Building resilience	Transforming vulnerability into skills	Dramatic activities developed self-awareness and coping abilities in challenging situations.
	Developing coping strategies	Children enhanced adaptability and resilience through experiential artistic activities.

**Table 4 children-12-01498-t004:** Superordinate and emergent meaning patterns reflecting communication and expressive development.

Emergent Themes	Illustrative Meaning Units	Interpretative Description
Cultivation of Oral language	Improvement of oral expression	Children developed clearer and more structured speech.
	Development of narrative skills	Strengthened ability to narrate experiences and emotions.
Externalization and management of emotions	Verbal expression of feelings	Children spoke more directly about their emotions.
	Symbolic management of emotions	Use of roles and stories to process and express emotions.
Dialogue and interaction	Active participation in discussions	Increased engagement in group dialogue and discussions.
	Expansion of interpersonal communication	Greater collaboration and communication with peers and teachers.
Strengthening Communicative self-confidence	Overcoming hesitation in expression	Reduced reluctance and improved courage in communication.
	Public presentation	Children shared stories or performances before others, reinforcing self-confidence.

**Table 5 children-12-01498-t005:** Superordinate and emergent meaning patterns reflecting trust and cooperation within the school context.

Emergent Themes	Illustrative Meaning Units	Interpretative Description
Group identity and cohesion	Shared goal achievement	Completing collective activities reinforced the sense of joint effort.
	Mutual support	Active assistance and encouragement among children during presentations and group tasks.
Trust and open communication	Sincere sharing of experiences	Children felt safe to express themselves, sharing personal thoughts and emotions within the group.
	Acceptance of diversity	Recognition and acceptance of diverse opinions, roles, and traits among peers.
Social networks and bonds	Creation of new friendships	Development of new interpersonal relationships that continue to this day.
	Cooperation beyond the program	Sustained communication and collaboration among students, even beyond the intervention.
Solidarity in Difficult circumstances	Support during difficulties	Practical and emotional support for peers facing challenges during the period of isolation.
	Collective problem-solving	Joint effort to overcome difficulties arising during theatrical activities and group work.

**Table 6 children-12-01498-t006:** Superordinate and emergent meaning patterns illustrating adaptive responses to new circumstances.

Emergent Themes	Illustrative Meaning Units	Interpretative Description
Managing change	Familiarity with the unpredictable	Children developed skills in accepting sudden changes in the program and daily routines.
	Coping with technical difficulties	Strengthened ability to handle technical problems and adapt to setbacks without anxiety.
Self-regulation	Emotional control	The intervention contributed to self-regulation and to managing negative emotions under difficult conditions.
	Initiative-taking	Children were encouraged to take initiative in solving problems without giving up easily.
Psychological flexibility	Role adaptation	Through alternating theatrical roles, children became more familiar with adopting different perspectives and responsibilities.
	Development of alternative strategies	Strengthened ability to find diverse ways of solving problems during activities.
Dealing with uncertainty	Tolerance of uncertainty	Systematic exposure to uncertain conditions during the intervention helped reduce fear of the unknown.
	Maintaining functionality	Despite changes and disruptions, children preserved their interest and active participation.

**Table 7 children-12-01498-t007:** Superordinate and emergent meaning patterns illustrating symbolic meaning-making and experiential interpretation through performing arts.

Emergent Themes	Illustrative Meaning Units	Interpretative Description
Symbolic expression of experiences	Theatrical metaphors	Use of theatrical symbols to represent crisis experiences.
	Storytelling/fairytale	Processing experiences through structured narratives.
Cultivation of hope and transcendence	Expression of expectations	Envisioning positive outcomes through roles and dramatic play.
	Creation of “pathways”	Overcoming difficulties by creating symbolic escapes.
Connecting personal experience with collective narrative	Personal stories on stage	Incorporating personal experiences into collective narratives.
	Identification with characters	Projecting personal concerns onto symbolic heroes.
Utilization of artistic forms at political/educational level	Transfer of good practices	Adapting theatrical practices for educational policy.
	Dissemination of artistic expressions	Promoting drama pedagogy within schools.

## Data Availability

The data presented in this study are available on request from the corresponding author.

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
