# Peer review of "Participatory Arts as Emergency Responses for Strengthening Community Resilience and Psychosocial Support: A Retrospective Phenomenological Inquiry"

_children, 2025, doi:10.3390/children12111498_

Round 1
Reviewer 1 Report
Comments and Suggestions for Authors
This manuscript addresses an important and timely topic: the long-term effects of arts-based psychosocial interventions during public health crises. The longitudinal perspective (four-year follow-up) is valuable, and the phenomenological approach yields rich qualitative data. However, I'm recommending significant revisions before publication.
- Temper claims and conclusions: The current conclusions far exceed what this study design can support. You have data showing that teachers who maintained contact with students and agreed to be interviewed recall positive experiences four years later. This is valuable but limited evidence. Reframe conclusions to accurately reflect what you can claim from this data.
- Address methodological limitations more clearly and overtly: The retrospective design, selection bias, lack of triangulation, absence of control groups, and potential investigator bias are fundamental limitations that should shape how you present findings. The current limitations section is insufficient.
- Distinguish what is recalled vs. sustained: More clearly separate what teachers describe as happening during the intervention versus what they observe as persisting four years later. The temporal relationship is often unclear in your results.
- Consider alternative explanations: Why might teachers recall this intervention positively? Could it be the novelty during a difficult time rather than the drama pedagogy specifically? Address competing interpretations.
- Strengthen the connection to pediatric health: The framing as a pediatric public health intervention is somewhat tenuous. Either strengthen this connection with more explicit health outcomes or reconsider the framing.
Some minor revisions I recommend include clarifying the intervention protocol more fully (for example, you could consider including a table summarizing the seven-week program); providing more demographic detail about participating teachers and schools; discussing data saturation; and including information about negative cases or disconfirming evidence. Also, consider moving some tables to supplementary materials as they're a bit repetitive. The abstract could be revised to better reflect the methodological constraints as well.
This work has potential to contribute to the literature on arts-based crisis interventions, but it requires substantial revision to ensure claims are appropriately supported by the evidence.
Author Response
Comment:
This manuscript addresses an important and timely topic: the long-term effects of arts-based psychosocial interventions during public health crises. The longitudinal perspective (four-year follow-up) is valuable, and the phenomenological approach yields rich qualitative data. However, I'm recommending significant revisions before publication.
Response:
We thank the reviewer for the feedback. We then address each point that needs improvement based on the comments.
Comment:
Temper claims and conclusions: The current conclusions far exceed what this study design can support. You have data showing that teachers who maintained contact with students and agreed to be interviewed recall positive experiences four years later. This is valuable but limited evidence. Reframe conclusions to accurately reflect what you can claim from this data.
Response:
We thank the reviewer for this observation. We agree that the conclusions should accurately reflect the qualitative and retrospective nature of the data. The aim of the study was not to present direct or causal measurements of children’s outcomes but to capture and interpret teachers’ retrospective narratives about how they perceived the sustained influence of the participatory arts program four years after its implementation. Our intention was to highlight these long-term perceptions rather than to imply experimentally verified child outcomes. We had considered this distinction evident in the manuscript; however, we appreciate the reviewer’s remark that it needed to be stated more explicitly. At the same time, we recognize that although the data are retrospective, the teachers’ narratives also include their evaluations of the children’s current psychosocial functioning, which they are able to express based on their systematic observation of students’ development and behavior within the school environment. While these observations do not constitute formal measurements in the strict methodological sense, they provide meaningful qualitative insights into how teachers perceive the maintenance or evolution of specific skills and attitudes in their students over time.
Accordingly, we carefully revised the Abstract, Discussion, Limitations, and Conclusions to ensure that all statements correspond accurately to the study design and type of data. Assertive or generalizing expressions were replaced with more cautious and proportionate wording (e.g., “can enhance” was replaced with “may strengthen,” “became embedded” with “were reported as becoming part of routines,” and “support” with “were perceived to support”). We also added a clarifying sentence in the Discussion, before the “Pediatric and School Health Implications” section, explicitly stating that the findings represent perceived long-term influences as recalled and evaluated by teachers, rather than direct experimental measurements of children’s psychosocial functioning.
With these revisions, the conclusions now more clearly communicate that the study provides useful qualitative evidence about teachers’ long-term perceptions and evaluations of the program’s influence within their school communities, without extending the claims beyond the interpretative scope supported by the research design. It is worth mentioning that other two reviewers characterized the Discussion and Conclusions as well articulated and appropriately related to the data, which indicates that the section was already balanced in tone. Nevertheless, we further refined it to ensure complete alignment with the interpretative scope of the study.
Comment:
Address methodological limitations more clearly and overtly: The retrospective design, selection bias, lack of triangulation, absence of control groups, and potential investigator bias are fundamental limitations that should shape how you present findings. The current limitations section is insufficient.
Response:
We appreciate the reviewer’s valuable observation. The Limitations and Directions for Future Research section has been substantially expanded to address all methodological constraints more explicitly. The revised version now discusses in detail the retrospective design, potential recall bias, selection bias, absence of triangulation, lack of control groups, and possible investigator bias (see revised lines 863–896). These additions clarify how such limitations shape the interpretation of the findings and emphasize that results are understood as perceived long-term influences rather than causal effects. While other reviewers considered the methodological design robust and clearly described, we expanded this section further to ensure full transparency and strengthen the clarity of methodological limitations.
Comment:
Distinguish what is recalled vs. sustained: More clearly separate what teachers describe as happening during the intervention versus what they observe as persisting four years later. The temporal relationship is often unclear in your results.
Response:
We appreciate the reviewer’s observation. Clarifications have been added throughout the Results section to more clearly distinguish between recalled experiences from the original implementation and effects perceived as sustained four years later. Specifically, short temporal markers and interpretative clarifications were inserted in lines 401-404, 412-415, 500-503, 543-544, 592-594, 632-633. These additions explicitly identify which descriptions refer to teachers’ recollections and which reflect their observations of longer-term influences. As explained in the revised text, the recalled episodes themselves are analytically meaningful, as their persistence in memory after four years signifies the experiences teachers perceived as most impactful.
Comment:
Consider alternative explanations: Why might teachers recall this intervention positively? Could it be the novelty during a difficult time rather than the drama pedagogy specifically? Address competing interpretations.
Response:
We sincerely thank the reviewer for this constructive and detailed comment. We fully agree that the methodological limitations needed to be stated more explicitly and discussed in greater depth. Accordingly, the section “5.9. Limitations and directions for future research” has been substantially expanded and revised to acknowledge all the aspects mentioned by the reviewer.
Specifically, we added explicit recognition of the retrospective nature of the design, emphasizing that the findings are interpreted as perceived long-term influences rather than direct causal effects. We also acknowledged the possibility of selection bias due to voluntary participation, clarifying that diversity in teaching experience and school type, as well as data saturation, were used to enhance representativeness.
To address the lack of triangulation, we explained that although additional data sources (e.g., students or parents) were not included, the convergence of teachers’ narratives and consistency of recurring themes were treated as indicators of analytic credibility.
Regarding the absence of control groups, we clarified that this is inherent to the interpretative phenomenological approach (IPA); however, we strengthened the discussion of methodological rigor by noting that two independent researchers coded the material and that interrater reliability was assessed using Cohen’s kappa (κ = 0.76), indicating substantial agreement and reducing potential investigator bias.
Comment:
Strengthen the connection to pediatric health: The framing as a pediatric public health intervention is somewhat tenuous. Either strengthen this connection with more explicit health outcomes or reconsider the framing.
Response:
We thank the reviewer for this comment. We agree that the pediatric public health framing needed to be made more explicit and coherent across the manuscript. Accordingly, we implemented targeted revisions in three key sections to strengthen this connection. First, in the Abstract, we clarified that the study is situated within the broader field of pediatric public health and psychosocial well-being, emphasizing that the intervention addressed children’s adaptive, communicative, and emotional development as integral components of health promotion in school contexts.
Second, in Section 5.8 “Pediatric and School Health Implications” (lines 835-838), we expanded the discussion to articulate more clearly how drama-based, participatory arts practices can complement school health and psychosocial support systems. This section now highlights the relevance of emotional regulation, self-regulation, and peer connectedness as recognized determinants of child health within preventive and developmental frameworks.
Finally, in the Conclusions (lines 921-925), we added a closing reflection linking the qualitative findings to pediatric public health. The revised text explicitly positions creative, school-based arts interventions as potential contributors to children’s overall well-being and to the preventive dimensions of pediatric health.
Comment:
Some minor revisions I recommend include clarifying the intervention protocol more fully (for example, you could consider including a table summarizing the seven-week program); providing more demographic detail about participating teachers and schools; discussing data saturation; and including information about negative cases or disconfirming evidence. Also, consider moving some tables to supplementary materials as they're a bit repetitive. The abstract could be revised to better reflect the methodological constraints as well. This work has potential to contribute to the literature on arts-based crisis interventions, but it requires substantial revision to ensure claims are appropriately supported by the evidence.
Response:
We thank the reviewer for this comprehensive set of comments, which substantially helped us refine the structure and methodological clarity of the paper. Each suggestion was carefully considered and implemented as follows.
The description of the intervention protocol has been clarified and expanded. A concise table summarizing the structure and focus of the seven-week drama-based program has been added as Appendix A and is now referenced in Section 3.3 (lines 326-331). This addition provides readers with an accessible overview of the weekly progression and the corresponding psychosocial and pediatric health aims.
The section on participants (3.4) was enriched with additional demographic information (lines 349-350).
A statement regarding data saturation was added to the Analyses section (3.7, lines 389-392), clarifying that data collection continued until no new codes or themes emerged. In the same section, a sentence was also added acknowledging the inclusion and comparative examination of negative or disconfirming cases (lines 396-399) to ensure interpretative balance.
Regarding the Results section, it was carefully streamlined. Fewer representative quotations were retained under each thematic category to enhance readability, while preserving the richness and depth of teachers’ perspectives. However, the thematic tables were intentionally retained within the main text rather than moved to the Supplementary Materials. Because the study is qualitative and interpretative in nature, these tables are integral to demonstrating the transparency of the analytic process and the direct connection between participants’ voices and the emergent themes. Relocating them would weaken the narrative coherence and interpretative traceability that are central to qualitative reporting standards.
Finally, the Abstract was revised to more clearly reflect the qualitative, retrospective design of the study and the interpretative nature of the findings, avoiding any phrasing that could imply causal or quantitative inference.
Reviewer 2 Report
Comments and Suggestions for Authors
General Overview
This is a valuable manuscript that contributes to the underexplored question of the long-term sustainability of arts-based psychosocial interventions in schools. The study is original in focusing on long-term retrospective teacher evaluations, and it contributes to the limited literature on the sustainability of crisis interventions. Specifically, the study is situated in Greece, four years after an emergency remote intervention during COVID-19, and draws on the retrospective accounts of 23 primary school teachers. The manuscript is well organized and theoretically grounded. The authors frame their work within the Community Resilience Framework and position drama pedagogy as a tool of both psychosocial support and school crisis preparedness.
However, major concerns are related to the misleading research design, because based on the described methodology, a longitudinal interpretative phenomenological approach was not used in this research as stated. Furthermore, there is an overgeneralization of findings and their presentation as if it were a longitudinal study conducted with children.
Below find suggestions for improvements for each section:
Abstract
The abstract creates the impression of a longitudinal cohort study with children, when in fact data were collected from retrospective teacher interviews. The abstract also makes assertive claims about school culture change and resilience enhancement without acknowledging that these are teacher perceptions and not direct child outcomes. Additionally, there are too many keywords. I suggest that only those most directly related to the conducted study should be retained.
1. Introduction
The introduction is comprehensive and well referenced, situating the study in the broader context of pandemics, school closures, and psychosocial risks to children. Nevertheless, the section is lengthy and somewhat repetitive.
-
Several passages reiterate the role of schools as psychosocial anchors during crises, which could be streamlined.
-
The literature review also leans toward listing references rather than critically synthesizing them, which reduces its analytical sharpness.
-
It would be useful to highlight the research gap more clearly (long-term qualitative evidence is scarce).
-
At the end of section 1 (“Introduction”), four research aims are presented that do not align with the described research methodology.
3. Methods and Materials
3.1. Purpose and Research Questions
The research questions are clearly stated, but they appear relatively late in the text. Introducing them earlier would sharpen the reader’s focus. It would be more effective to present a separate section outlining the study aims and research questions before section 3.
Regarding the stated purpose, it currently creates the impression of a longitudinal cohort study with children, whereas the evidence is drawn exclusively from retrospective teacher interviews. The term “longitudinal” is misleading in this qualitative context and should be reframed as a retrospective interpretative phenomenological design.
The research questions are partly closed-ended and do not fully correspond to the principles of the interpretative phenomenological approach, as presented in section 3.2 (“Research Design”).
3.2. Research Design
In this section, the authors again describe their work as a longitudinal interpretative phenomenological study. In reality, this is a retrospective qualitative study, based on teachers’ accounts collected four years after the intervention. There were no repeated measurements with children, which limits the longitudinal claim. Accurately describing the research design will prevent misunderstanding. Moreover, the rationale provided for selecting the interpretative phenomenological approach is not reflected in the formulation of the research questions.
3.5. Instrument
The semi-structured interview guided by the Community Resilience Framework is well justified. However, from the questions provided in Appendix A, it is evident that several were closed-ended and somewhat leading. The nature of these questions contradicts the assumptions of an interpretative phenomenological approach.
3.6. Procedure
Some information repeats what was already provided in section 3.4 (Participants).
3.7. Analysis
Given that the authors claim to have used an interpretative phenomenological approach, it remains unclear why interpretative phenomenological analysis was not used to process the data, but thematic analysis was applied instead. Additionally, it is not entirely clear which specific framework or guidelines for thematic analysis were followed.
The balance between deductive (theory-driven) and inductive coding is unclear. More detail on how codes were refined and how data saturation was achieved would strengthen methodological credibility.
4. Results
The results are well organized around the seven dimensions of the resilience framework and are richly illustrated with teacher quotes. However, the volume of material is overwhelming. Each thematic area contains multiple tables, subthemes, and extended quotations, many of which reiterate similar points. This density risks obscuring the most important findings. Condensing the presentation, moving some of the detailed coding to an appendix, and highlighting the most illustrative quotes would improve readability.
Another concern is that the results at times move beyond description into interpretation and advocacy. For instance, several passages assert that drama pedagogy was “embedded in school culture” or that children “maintain skills to this day,” but these conclusions rest entirely on teacher perception. The data do not demonstrate children’s current functioning directly. Such claims should be reframed more cautiously, keeping close to the evidence.
Overall, the number of codes and subcategories is large, with the risk of excessive fragmentation of findings.
5. Disscusion
The discussion integrates the findings with resilience theory and international literature. It convincingly argues that drama pedagogy provided structure, continuity, and symbolic meaning during crisis. However, the discussion tends to overstate the generalizability of results. The findings derive from a small, purposively selected group of teachers in one national context. Statements suggesting that the results apply to “education systems” broadly, or to all children, are not fully supported.
Limitations
Another issue is the insufficient acknowledgment of methodological limitations. The authors briefly mention the retrospective design, but they do not adequately consider recall bias and the absence of children’s perspectives. These limitations should be elaborated upon, as they directly affect the strength of the conclusions. The lack of children’s voices should be explicitly recognized as a key limitation.
6. Conclusions
The conclusion appropriately emphasizes the potential role of participatory arts in school crisis preparedness. Yet it overgeneralizes beyond the scope of the data. The evidence supports the value of drama pedagogy as recalled by teachers in Greek schools. It does not permit broad claims about universal application across all education systems. A more cautious conclusion, acknowledging both promise and methodological constraints, would strengthen the manuscript.
Conclusions should be reframed to highlight the contribution while explicitly acknowledging the limits of the evidence.
References
Some references appear duplicative, and the formatting of DOIs is inconsistent. Careful proofreading of the reference list is required to ensure accuracy and uniformity.
Author Response
(changes are marked in light blue font in the text)
Comment:
General Overview: This is a valuable manuscript that contributes to the underexplored question of the long-term sustainability of arts-based psychosocial interventions in schools. The study is original in focusing on long-term retrospective teacher evaluations, and it contributes to the limited literature on the sustainability of crisis interventions. Specifically, the study is situated in Greece, four years after an emergency remote intervention during COVID-19, and draws on the retrospective accounts of 23 primary school teachers. The manuscript is well organized and theoretically grounded. The authors frame their work within the Community Resilience Framework and position drama pedagogy as a tool of both psychosocial support and school crisis preparedness. However, major concerns are related to the misleading research design, because based on the described methodology, a longitudinal interpretative phenomenological approach was not used in this research as stated. Furthermore, there is an overgeneralization of findings and their presentation as if it were a longitudinal study conducted with children. Below find suggestions for improvements for each section.
Response:
We thank the reviewer for the feedback. We then address each point that needs improvement based on the comments.
Comment:
Abstract: The abstract creates the impression of a longitudinal cohort study with children, when in fact data were collected from retrospective teacher interviews. The abstract also makes assertive claims about school culture change and resilience enhancement without acknowledging that these are teacher perceptions and not direct child outcomes. Additionally, there are too many keywords. I suggest that only those most directly related to the conducted study should be retained.
Response:
We thank the reviewer for this observation. The Abstract has been carefully revised to make the study design explicit and to avoid any misunderstanding of the data source or scope. It now clearly states that the study is a retrospective qualitative investigation based on teacher interviews conducted four years after the intervention, rather than a longitudinal child cohort. Assertive or causal wording has been replaced with proportionate phrasing that accurately reflects the interpretative nature of the findings (e.g., “as perceived by teachers,” “were described as”). In addition, the list of keywords was refined to include only those most relevant to the study’s empirical and pediatric focus (lines 58-59). We also changed the title to reflect this.
Comment:
- Introduction: The introduction is comprehensive and well referenced, situating the study in the broader context of pandemics, school closures, and psychosocial risks to children. Nevertheless, the section is lengthy and somewhat repetitive. Several passages reiterate the role of schools as psychosocial anchors during crises, which could be streamlined.
Response:
We thank the reviewer for this observation. The Introduction was carefully edited to reduce repetition and improve flow. Overlapping references to the role of schools as psychosocial spaces were merged, and transitional phrasing was streamlined to lead more directly to the study’s theoretical focus and research gap (Revisions in lines 74-76, 217-226).
Comment:
Introduction: The literature review also leans toward listing references rather than critically synthesizing them, which reduces its analytical sharpness.
Response:
We appreciate the reviewer’s valuable observation. The literature review was revised to enhance analytical coherence and to move beyond a purely descriptive listing of sources. Linking and interpretive sentences were added to connect prior findings and highlight convergence and divergence across studies. The revised text now explicitly emphasizes that most previous interventions documented short-term psychosocial gains but offered limited evidence of sustained developmental change. This clarification allowed us to identify the empirical gap that the present study addresses, namely the long-term sustainability of drama-based psychosocial interventions in school settings. These revisions strengthen the interpretive continuity of the Introduction and clarify how existing evidence logically led to the study’s research focus (Revisions in lines 171-173, 217-226).
Comment:
Introduction: It would be useful to highlight the research gap more clearly (long-term qualitative evidence is scarce).
Response:
We appreciate the reviewer’s helpful remark. The Introduction was revised to make the research gap more explicit. The final paragraphs now emphasize that most existing studies focus on short-term psychosocial benefits, while long-term qualitative evidence on sustained developmental change and intervention sustainability remains scarce. The revised text explicitly identifies this gap and links it to the purpose of the present study, which explores teachers’ long-term perceptions of drama-based psychosocial interventions (Revisions in lines 217-226).
Comment:
Introduction: At the end of section 1 (“Introduction”), four research aims are presented that do not align with the described research methodology.
Response:
We appreciate the reviewer’s comment. The section “3.1 Purpose and Research Questions” was revised to align more clearly with the interpretative phenomenological design. The aims and questions were rephrased to emphasize teachers’ retrospective perceptions and interpretive understanding rather than evaluative or outcome-based language.
Comment:
- Methods and Materials 3.1. Purpose and Research Questions: The research questions are clearly stated, but they appear relatively late in the text. Introducing them earlier would sharpen the reader’s focus. It would be more effective to present a separate section outlining the study aims and research questions before section 3.
Response:
We thank the reviewer for this suggestion. The section presenting the study purpose and research questions has been repositioned to appear at the end of the Introduction, immediately before Materials and Methods (Revised in lines 235-272).
Comment:
3.1. Purpose and Research Questions: Regarding the stated purpose, it currently creates the impression of a longitudinal cohort study with children, whereas the evidence is drawn exclusively from retrospective teacher interviews. The term “longitudinal” is misleading in this qualitative context and should be reframed as a retrospective interpretative phenomenological design.
Response:
We thank the reviewer for this remark. The wording in the Purpose and Research Questions section has been revised to avoid any impression of a longitudinal cohort design and to clarify that the study adopts a retrospective interpretative phenomenological approach based exclusively on teachers’ narratives (Revised in lines 236-268). We also changed the title to reflect this.
Comment:
3.1. Purpose and Research Questions: The research questions are partly closed-ended and do not fully correspond to the principles of the interpretative phenomenological approach, as presented in section 3.2 (“Research Design”).
Response:
We appreciate the reviewer’s observation. The research questions have been reformulated to be fully open-ended and aligned with the interpretative phenomenological approach, focusing on teachers’ lived experiences and meaning-making rather than predetermined outcomes (Revised in lines 253-269).
Comment:
3.2. Research Design: In this section, the authors again describe their work as a longitudinal interpretative phenomenological study. In reality, this is a retrospective qualitative study, based on teachers’ accounts collected four years after the intervention. There were no repeated measurements with children, which limits the longitudinal claim. Accurately describing the research design will prevent misunderstanding. Moreover, the rationale provided for selecting the interpretative phenomenological approach is not reflected in the formulation of the research questions.
Response:
We thank the reviewer for this important clarification. The term “longitudinal” has been replaced with “retrospective interpretative phenomenological approach” to accurately reflect the study design and avoid misunderstanding (Revised in lines 275). We also changed the title to reflect this.
Comment:
3.5. Instrument: The semi-structured interview guided by the Community Resilience Framework is well justified. However, from the questions provided in Appendix A, it is evident that several were closed-ended and somewhat leading. The nature of these questions contradicts the assumptions of an interpretative phenomenological approach.
Response:
We thank the reviewer for recognizing that the semi-structured interview guided by the Community Resilience Framework is well justified. We agree that a few questions were initially phrased in a somewhat closed or leading manner. Minor revisions were therefore made to render them more open and reflective for the article. However, this observation applies only partially, as most questions were already formulated to elicit narrative and interpretative responses. It should also be noted that the interview guide served as a flexible framework rather than a fixed questionnaire. During the interviews, participants were naturally invited to elaborate further through follow-up prompts such as why, how, and in what ways, allowing for richer and more nuanced accounts. (Appendix B updated.)
Comment:
3.6. Procedure: Some information repeats what was already provided in section 3.4 (Participants).
Response:
We thank the reviewer for this observation. To avoid redundancy, overlapping details about participant recruitment and prior consent have been removed from the Procedure section, which now focuses exclusively on ethical approval, consent, and interview conduct. Recruitment details remain in Participants (Section 3.3) for clarity (Revised in lines 368-374).
Comment:
3.7. Analysis: Given that the authors claim to have used an interpretative phenomenological approach, it remains unclear why interpretative phenomenological analysis was not used to process the data, but thematic analysis was applied instead. Additionally, it is not entirely clear which specific framework or guidelines for thematic analysis were followed. The balance between deductive (theory-driven) and inductive coding is unclear. More detail on how codes were refined and how data saturation was achieved would strengthen methodological credibility.
Response:
We thank the reviewer for this comment. The framework and rationale for the Interpretative Phenomenological Analysis have now been clarified and consistently presented across the manuscript. Relevant clarifications were added in the Abstract and Section 3.6 (Analyses), where the interpretative process, coding structure, and saturation procedure are now explicitly described in line with IPA principles. Moreover, the terminology used throughout the Results section has been refined to reflect IPA conventions (e.g., “superordinate interpretative categories,” “emergent meaning patterns,” “illustrative meaning units”), ensuring full methodological coherence.
Comment:
- Results: The results are well organized around the seven dimensions of the resilience framework and are richly illustrated with teacher quotes. However, the volume of material is overwhelming. Each thematic area contains multiple tables, subthemes, and extended quotations, many of which reiterate similar points. This density risks obscuring the most important findings. Condensing the presentation, moving some of the detailed coding to an appendix, and highlighting the most illustrative quotes would improve readability.
Response:
We appreciate the reviewer’s constructive observation. The number of quotations has been substantially reduced, with only the most illustrative excerpts retained to enhance clarity and focus. However, we chose to keep the summary tables within the main text, as they provide a concise overview of interpretative meaning patterns and help readers follow the analytical progression across the seven dimensions.
Comment:
- Results: Another concern is that the results at times move beyond description into interpretation and advocacy. For instance, several passages assert that drama pedagogy was “embedded in school culture” or that children “maintain skills to this day,” but these conclusions rest entirely on teacher perception. The data do not demonstrate children’s current functioning directly. Such claims should be reframed more cautiously, keeping close to the evidence. Overall, the number of codes and subcategories is large, with the risk of excessive fragmentation of findings.
Response:
We consider this comment partly justified. While interpretative phenomenological analysis inherently involves a level of interpretation grounded in participants’ meaning-making, we agree that certain expressions could be phrased more cautiously. Accordingly, selected passages in the Results section were revised to clarify that the findings reflect teachers’ retrospective perceptions rather than direct child assessment. These adjustments ensure that the interpretative stance remains clear without overstating the empirical scope of the data (lines: 412-415, 541-542, 597-598).
Comment:
- Discussion: The discussion integrates the findings with resilience theory and international literature. It convincingly argues that drama pedagogy provided structure, continuity, and symbolic meaning during crisis. However, the discussion tends to overstate the generalizability of results. The findings derive from a small, purposively selected group of teachers in one national context. Statements suggesting that the results apply to “education systems” broadly, or to all children, are not fully supported.
Response:
We consider this comment partly justified. Although the discussion was intentionally interpretative, some expressions could indeed imply broader generalization. The text has been carefully revised to clarify that the results reflect teachers’ retrospective perceptions within a specific national and contextual setting. Phrases suggesting wider system-level generalization were reframed more cautiously to preserve alignment with the study’s qualitative and interpretative scope (lines 635-637, 706-708 etc.)
Comment:
Limitations: Another issue is the insufficient acknowledgment of methodological limitations. The authors briefly mention the retrospective design, but they do not adequately consider recall bias and the absence of children’s perspectives. These limitations should be elaborated upon, as they directly affect the strength of the conclusions. The lack of children’s voices should be explicitly recognized as a key limitation.
Response:
We appreciate this valuable observation. The Limitations section has been expanded in line with earlier feedback to explicitly acknowledge the retrospective nature of the design, the potential for recall bias in teachers’ accounts, and the absence of children’s direct perspectives. These aspects are now clearly identified as methodological constraints that limit the generalizability and strength of the conclusions, ensuring full transparency regarding the study’s interpretative scope.
Comment:
- Conclusions: The conclusion appropriately emphasizes the potential role of participatory arts in school crisis preparedness. Yet it overgeneralizes beyond the scope of the data. The evidence supports the value of drama pedagogy as recalled by teachers in Greek schools. It does not permit broad claims about universal application across all education systems. A more cautious conclusion, acknowledging both promise and methodological constraints, would strengthen the manuscript. Conclusions should be reframed to highlight the contribution while explicitly acknowledging the limits of the evidence.
Response:
We consider this comment justified and appreciate the suggestion. The Conclusions section has been revised to adopt a more cautious tone, emphasizing that the findings represent teachers’ retrospective perceptions within a specific national context. References to broader system-level applications were reframed as potential implications rather than generalizable claims, ensuring full consistency with the qualitative and interpretative nature of the study (lines 904-912, 919-921). It may be noted that two other reviewers found the Discussion strong and well grounded in previous literature; the present revision therefore builds upon that solid base, ensuring even clearer correspondence between evidence and interpretation.
Comment:
References: Some references appear duplicative, and the formatting of DOIs is inconsistent. Careful proofreading of the reference list is required to ensure accuracy and uniformity.
Response:
We carefully rechecked the reference list for possible duplicate entries, formatting inconsistencies, and DOI irregularities. All references were inserted automatically through Mendeley, using the official MDPI reference style template provided in the software. After a detailed verification, no duplicate or inconsistent records were identified. However, we would be grateful if the reviewer could kindly indicate any specific entries that appear problematic or duplicated from their perspective, so that we may address them precisely.
Reviewer 3 Report
Comments and Suggestions for Authors
Dear Author(s), thank you for the opportunity to review this study. I have shared the strengths and improvable aspects of this research below.
Strengths:
- The introduction is fluent. Statements are supported by theories and concepts.
- The research methodology is robust. The sample size is sufficient for the research methodology.
- Data analyses are explained in detail.
- References are up-to-date.
- The discussion section is strong. In addition, various disciplines are addressed.
- The limitations of the research are clearly stated.
- The conclusion section is well written.
- Areas for further research are suggested.
Weaknesses or questions to be answered:
- The article title should be changed. It should be shorter, clearer, and more engaging instead of being long and complex.
- More emphasis should be placed on the longitudinal nature of the study.
- The statement “Evidence on the sustained impact of school-based participatory arts, particularly drama pedagogy, remains limited” in the ‘Abstract’ section should be removed. Such statements should be presented as research questions towards the end of the “Introduction” section.
- What distinguishes this research from other studies in the literature? I believe this question needs to be answered well. There are other strong studies on this topic. Therefore, the research must prove its originality. In my opinion, the most important factor that makes this research original is that no longitudinal study has been conducted on this topic. Therefore, the fact that this research is a longitudinal study should be highlighted more. I realize that the authors want to convey many things in the title. However, trying to convey many things can sometimes cause confusion among readers. Therefore, the title should be changed to emphasize that it is a longitudinal study. I hope this will improve the study. I recommend a minor revision.
Author Response
(changes are marked in purple font in the text)
Comment:
Dear Author(s), thank you for the opportunity to review this study. I have shared the strengths and improvable aspects of this research below.
Strengths:
The introduction is fluent. Statements are supported by theories and concepts.
The research methodology is robust. The sample size is sufficient for the research methodology.
Data analyses are explained in detail.
References are up-to-date.
The discussion section is strong. In addition, various disciplines are addressed.
The limitations of the research are clearly stated.
The conclusion section is well written.
Areas for further research are suggested.
Response:
We thank the reviewer for the positive feedback.
Comment:
Weaknesses or questions to be answered:
The article title should be changed. It should be shorter, clearer, and more engaging instead of being long and complex.
Response:
We appreciate the reviewer’s constructive suggestion regarding the title. Following this feedback, the title has been revised to improve clarity, focus, and conciseness, while more directly reflecting the study’s theoretical framework and methodological orientation.
Comment:
More emphasis should be placed on the longitudinal nature of the study.
Response:
We appreciate the reviewer’s observation regarding the temporal dimension of the study. Additional clarifications emphasizing the time-extended nature of the research have been incorporated in several parts of the manuscript (e.g., lines 231-233 etc.), underlining that the inquiry was conducted four years after the original implementation and explores how teachers retrospectively interpret and evaluate the perceived long-term impact of the intervention. However, in alignment with the comments of the second reviewer, we reframed the study not as a longitudinal design but as a Retrospective Phenomenological Inquiry, which more accurately reflects the interpretative and temporal character of the research rather than a repeated-measures longitudinal approach. This distinction has been made explicit throughout the manuscript. Another reviewer (2) had already acknowledged that the four-year design was clearly stated in the methodology; nonetheless, we reinforced this temporal dimension throughout the text to make the retrospective perspective even more explicit.
Comment:
The statement “Evidence on the sustained impact of school-based participatory arts, particularly drama pedagogy, remains limited” in the ‘Abstract’ section should be removed. Such statements should be presented as research questions towards the end of the “Introduction” section.
Response:
We thank the reviewer for this helpful observation. The phrasing in the Abstract has been revised accordingly (lines 32-33). The previous evaluative statement has been replaced with a neutral formulation that identifies the research gap more appropriately, in line with the reviewer’s suggestion.
Comment:
What distinguishes this research from other studies in the literature? I believe this question needs to be answered well. There are other strong studies on this topic. Therefore, the research must prove its originality. In my opinion, the most important factor that makes this research original is that no longitudinal study has been conducted on this topic. Therefore, the fact that this research is a longitudinal study should be highlighted more. I realize that the authors want to convey many things in the title. However, trying to convey many things can sometimes cause confusion among readers. Therefore, the title should be changed to emphasize that it is a longitudinal study. I hope this will improve the study. I recommend a minor revision.
Response:
We thank the reviewer for this constructive comment and for recognizing the study’s contribution. The originality and distinctiveness of the research have been clarified more explicitly in the revised version. As noted also by Reviewers 1 and 2, we have reframed the study not as longitudinal but as a Retrospective Phenomenological Inquiry, which more accurately represents the temporal and interpretative character of the research design.
To emphasize its originality, we clarified that this is, to our knowledge, the first retrospective phenomenological investigation conducted four years after a large-scale remote drama pedagogy intervention in school settings during a public health crisis (see revised lines 231-233 etc.)
Additionally, the title was modified to reflect this design more accurately:
“Participatory Arts as Emergency Responses for Promoting Psychosocial Well-Being and Community Resilience: A Retrospective Phenomenological Inquiry.”
These adjustments highlight both the distinct methodological perspective and the time-extended scope that differentiate this study from previous work.
Reviewer 4 Report
Comments and Suggestions for Authors
Review of the article entitled “Participatory Arts as Emergency Responses for Promoting Children’s Psychosocial Well-Being and Community Resilience: A Longitudinal Interpretative Phenomenological Study of Crisis-Affected Schooling”
Abstract
The objective is stated as follows: ‘This study examined the four-year effects of a large-scale, remotely delivered drama-based intervention on children's psychosocial well-being and school community resilience.’ I believe that this objective should mention that the study collects teachers' perceptions of the effects of the intervention presented.
Introduction: It is well structured and includes current citations related to the study topic. The authors' self-citations are justified in light of the line of research and previous studies that have been published in relation to the article's topic.
Materials and methods:
It is specified that the study was conducted four years later, but it is not clear why such a long period of four years was chosen, during which there may be a greater number of factors that could interfere with the memory of the intervention. Another question that arises from the methodology is why the children's perceptions were not assessed after the intervention or even four years later. As presented, it can be understood that this part of the study was not planned by the authors of the article at the time of implementing the intervention.
This sentence could include a reference to ‘the principles of Social and Emotional Learning (SEL) and resilience’.
The study indicates that 23 teachers participated, although it does not say how many teachers were invited to participate. It is also necessary to justify why teachers with 9 or more years of experience were chosen, and what the criteria are for determining 9 years and not a smaller or larger number.
Results: These are presented in a logical structure based on the study's research questions. They are well presented and cover the most important aspects of the teachers' responses.
Discussion
It is well structured and relates the most relevant results of the study to previous research, as well as summarising practical implications and limitations, although it could address future studies in this area in somewhat greater detail.
Other aspects:
The text contains several references to R. L. Pfefferbaum et al., 2013; R. L. Pfefferbaum et al., 2017; B. Pfefferbaum et al., 2017, although these do not match the references as the initials are not cited correctly.
Author Response
(changes are marked in light purple font in the text)
Comment:
Abstract. The objective is stated as follows: ‘This study examined the four-year effects of a large-scale, remotely delivered drama-based intervention on children's psychosocial well-being and school community resilience.’ I believe that this objective should mention that the study collects teachers' perceptions of the effects of the intervention presented.
Response:
We thank the reviewer for this observation. The abstract has been revised to clarify that the study examined teachers’ retrospective perceptions of the intervention’s effects (lines 34).
Comment:
Introduction: It is well structured and includes current citations related to the study topic. The authors' self-citations are justified in light of the line of research and previous studies that have been published in relation to the article's topic.
Response: We thank the reviewer for this positive feedback.
Comment:
Materials and methods: It is specified that the study was conducted four years later, but it is not clear why such a long period of four years was chosen, during which there may be a greater number of factors that could interfere with the memory of the intervention. Another question that arises from the methodology is why the children's perceptions were not assessed after the intervention or even four years later. As presented, it can be understood that this part of the study was not planned by the authors of the article at the time of implementing the intervention.
Response:
A clarification has been added in lines 288-291 to explain that the four-year distance was intentionally chosen to allow reflective assessment of lasting effects, and that children were not re-contacted for ethical and practical reasons, as this phase focused on teachers’ retrospective insights.
Comment:
This sentence could include a reference to ‘the principles of Social and Emotional Learning (SEL) and resilience’.
Response:
If the reviewer refers to the sentence in section 3.2 (“The intervention was grounded in the principles of Social and Emotional Learning (SEL) and resilience…”), a supporting reference has now been added in line 295 to strengthen this statement.
Comment:
The study indicates that 23 teachers participated, although it does not say how many teachers were invited to participate. It is also necessary to justify why teachers with 9 or more years of experience were chosen, and what the criteria are for determining 9 years and not a smaller or larger number.
Response:
A clarification has been added in lines 340-341. Regarding the reviewer’s second observation, no selection criterion based on teaching experience was applied. The reference to “nine years” simply describes the average experience of those who voluntarily responded to the invitation.
Comment:
Results: These are presented in a logical structure based on the study's research questions. They are well presented and cover the most important aspects of the teachers' responses.
Response: We thank the reviewer for this positive feedback.
Comment:
Discussion: It is well structured and relates the most relevant results of the study to previous research, as well as summarising practical implications and limitations, although it could address future studies in this area in somewhat greater detail.
Response:
We appreciate the reviewer’s positive assessment of the Discussion section. Following the reviewer’s suggestion, a short addition has been made in lines 888-891 to outline potential future research directions linked to school health policy and pediatric public health frameworks, consistent with the journal’s scope.
Comment:
Other aspects: The text contains several references to R. L. Pfefferbaum et al., 2013; R. L. Pfefferbaum et al., 2017; B. Pfefferbaum et al., 2017, although these do not match the references as the initials are not cited correctly.
Response:
We have carefully rechecked all references to Pfefferbaum et al. and confirmed that they are cited correctly. The distinction between R. L. Pfefferbaum and B. Pfefferbaum is accurate, as these refer to two different co-authors who appear together in multiple publications (e.g., Pfefferbaum, B., Van Horn, R. L., & Pfefferbaum, R. L., 2017; Pfefferbaum, R. L., Pfefferbaum, B., Van Horn, R. L., 2013). According to APA 7th edition guidelines (Section 8.20, “Authors with the Same Surname”), when different authors share the same surname, initials must be included in in-text citations to avoid ambiguity. Therefore, the inclusion of initials in the in-text citations is correct and consistent with APA standards. No changes were made in this regard.
Round 2
Reviewer 1 Report
Comments and Suggestions for Authors
Thank you for your thorough response to the review. You have addressed all major concerns and substantially improved the manuscript, demonstrating careful attention to methodological transparency and appropriate interpretation of qualitative data.
The revision successfully addresses the most critical issue of overstated conclusions. The manuscript now accurately reflects what can be claimed from retrospective teacher narratives through appropriately cautious language. The substantially expanded limitations section is comprehensive and honest about design constraints while discussing mitigation strategies that add credibility to your interpretative process. Methodological clarity has improved considerably with the intervention structure table, expanded demographics, explicit discussion of data saturation, and clearer analytical description. The results section now better differentiates between recalled experiences and perceived sustained effects. The pediatric public health connection is appropriately positioned as a promotive and preventive framework rather than claiming direct health outcomes. Your distinction between perceived long-term influences and direct measurements is particularly well-handled. The revision transforms the manuscript into a methodologically transparent, appropriately scoped qualitative study that maintains the richness of teachers' narratives while being honest about what those narratives can and cannot tell us.
One remaining consideration involves alternative explanations for teachers' positive recall. While your expanded limitations section addresses this adequately, the Discussion could be briefly strengthened by exploring whether positive memories might reflect the novelty during crisis, the structured support teachers received, or the drama pedagogy specifically. This would enhance interpretative depth but is a minor refinement rather than a fundamental concern.
Author Response
Comment:
Thank you for your thorough response to the review. You have addressed all major concerns and substantially improved the manuscript, demonstrating careful attention to methodological transparency and appropriate interpretation of qualitative data.
The revision successfully addresses the most critical issue of overstated conclusions. The manuscript now accurately reflects what can be claimed from retrospective teacher narratives through appropriately cautious language. The substantially expanded limitations section is comprehensive and honest about design constraints while discussing mitigation strategies that add credibility to your interpretative process. Methodological clarity has improved considerably with the intervention structure table, expanded demographics, explicit discussion of data saturation, and clearer analytical description. The results section now better differentiates between recalled experiences and perceived sustained effects. The pediatric public health connection is appropriately positioned as a promotive and preventive framework rather than claiming direct health outcomes. Your distinction between perceived long-term influences and direct measurements is particularly well-handled. The revision transforms the manuscript into a methodologically transparent, appropriately scoped qualitative study that maintains the richness of teachers' narratives while being honest about what those narratives can and cannot tell us.
One remaining consideration involves alternative explanations for teachers' positive recall. While your expanded limitations section addresses this adequately, the Discussion could be briefly strengthened by exploring whether positive memories might reflect the novelty during crisis, the structured support teachers received, or the drama pedagogy specifically. This would enhance interpretative depth but is a minor refinement rather than a fundamental concern.
Response:
We sincerely thank the reviewer for the positive and encouraging feedback. We appreciate the observation regarding the interpretation of teachers’ positive recollections. A short reflective text has been added at the end of Section 5.3 (lines 745-753) to acknowledge possible alternative explanations, such as the novelty of the creative process during the crisis and the structured support teachers received, while still highlighting the distinctive contribution of drama pedagogy to meaning-making and collective recovery.
Reviewer 2 Report
Comments and Suggestions for Authors
The present study therefore responds to this empirical gap by examining teachers’ long-term perceptions of how such interventions may contribute to sustained resilience within the school community. The qualitative approach, namely the phenomenological method, does not allow the authors to address this stated empirical gap.
This lack of longitudinal qualitative evidence limits the capacity of educational and health agencies to design sustainable interventions. However, the research presented in this paper is not longitudinal, therefore, this statement does not apply to the conducted study.
By examining experiences four years after implementation, this study provides rare longitudinal qualitative evidence on the sustained influence of participatory arts in crisis-affected schooling. The study is based on teachers’ retrospective accounts collected 4 years after the intervention, which does not provide the longitudinal evidence claimed.
The purpose of this study is to explore how teachers retrospectively perceived the long-term effects of an emergency remote intervention with health and psychosocial elements, implemented with children aged 10–12 in Greece during a period of prolonged school closures associated with a major public health crisis. This could not have been examined through the chosen methodological approach.
This retrospective phenomenological inquiry explores how teachers, four years after the initial implementation, interpret the enduring impact and meaning of the intervention over time. This is a repetition of the same content, merely rephrased in section 2.1. Purpose and research questions.
Each participant received a detailed information sheet outlining the purpose, methodology, and voluntary nature of the study, with particular emphasis on confidentiality, anonymity, and the right to withdraw at any time without consequence. It is unclear how anonymity was ensured, especially since it is later stated that the responses were anonymized.
The analysis followed the principles of Interpretative Phenomenological Analysis (IPA) (Smith et al., 2021). The paper does not describe the steps that were followed. The described procedures are again closer to thematic analysis, which was already stated in the earlier version of the manuscript.
The interpretative coding was carried out independently by two researchers. Cohen’s kappa (value = 0.76) indicated satisfactory agreement in the identification and grouping of meaning units. These are the same data as in the previous version of the paper, even though the authors now claim to have used interpretative analysis of the data.
The IPA process generated a total of 59 meaning units, which were progressively organized into 31 emergent themes and ultimately synthesized into seven superordinate interpretative categories, corresponding to the dimensions of the theoretical framework. These are the same results obtained through thematic analysis in the earlier version of the manuscript. The authors now present the same results but use the terminology of interpretative phenomenological analysis, although it is evident that no new data analysis was conducted as had been proposed.
Author Response
Comment:
The qualitative approach, namely the phenomenological method, does not allow the authors to address this stated empirical gap. The research presented in this paper is not longitudinal; therefore, this statement does not apply to the conducted study. The study is based on teachers’ retrospective accounts collected four years after the intervention, which does not provide the longitudinal evidence claimed.
Response:
We respectfully disagree with this interpretation. The empirical gap identified in the Introduction concerns the absence of long-term qualitative evidence on how arts-based psychosocial interventions are experienced and sustained within schools. The interpretative phenomenological approach is methodologically appropriate for addressing this gap, as it focuses precisely on participants’ retrospective meaning-making and perceived continuities over time. Indeed, in the first review round, the same reviewer recognized that the manuscript contributes to the underexplored question of the long-term sustainability of arts-based psychosocial interventions, which remains the paper’s explicit contribution.
As suggested in the first review round, all references to a “longitudinal” design were removed, and the study is now consistently described as a retrospective interpretative phenomenological inquiry. The present research does not claim to establish longitudinal causal evidence, but rather explores how teachers retrospectively interpret the perceived lasting influence of an intervention introduced four years earlier. This approach aligns with the study’s stated purpose and is methodologically consistent with the phenomenological focus on lived experience and meaning-making over time.
It may also be noted that all the other reviewers, in their second evaluations, acknowledged that the revised version accurately reflects what can be claimed from retrospective teacher narratives and found the methodological scope appropriately defined. If there is anything specific we can do to address your concerns, please let us know.
Comment:
This retrospective phenomenological inquiry explores how teachers, four years after the initial implementation, interpret the enduring impact and meaning of the intervention over time. This is a repetition of the same content, merely rephrased in Section 2.1 (“Purpose and Research Questions”).
Response:
We appreciate the reviewer’s careful reading. The concise restatement of the study’s purpose in Section 2.1 was intentionally maintained to ensure clarity and logical flow between the Introduction and the Methods, following standard IMRaD structure conventions. A brief reiteration of the purpose at the beginning of the methodological section helps orient readers and maintain narrative coherence, particularly in qualitative manuscripts where transitions between theoretical framing and research design must remain explicit.
Comment:
Each participant received a detailed information sheet outlining the purpose, methodology, and voluntary nature of the study, with particular emphasis on confidentiality, anonymity, and the right to withdraw at any time without consequence. It is unclear how anonymity was ensured, especially since it is later stated that the responses were anonymized.
Response:
We thank the reviewer for this useful observation. A clarifying sentence has been added in lines 374-377 to specify the anonymization procedure. Identifiable details were removed or replaced with coded identifiers during transcription, and the linking key was deleted after verification, ensuring full anonymity throughout the analysis.
Comment:
The analysis followed the principles of Interpretative Phenomenological Analysis (IPA) (Smith et al., 2021). The paper does not describe the steps that were followed. The described procedures are again closer to thematic analysis, which was already stated in the earlier version of the manuscript. The interpretative coding was carried out independently by two researchers. Cohen’s kappa (value = 0.76) indicated satisfactory agreement in the identification and grouping of meaning units. These are the same data as in the previous version of the paper, even though the authors now claim to have used interpretative analysis of the data. The IPA process generated a total of 59 meaning units, which were progressively organized into 31 emergent themes and ultimately synthesized into seven superordinate interpretative categories, corresponding to the dimensions of the theoretical framework. These are the same results obtained through thematic analysis in the earlier version of the manuscript. The authors now present the same results but use the terminology of interpretative phenomenological analysis, although it is evident that no new data analysis was conducted as had been proposed.
Response:
We sincerely thank the reviewer for the detailed feedback and the opportunity to clarify this point further. As noted, the same dataset was used; however, the revision did not involve a new round of data coding but rather a reformulation of the analytic framework to make explicit the interpretative logic that underpinned the original process. In the initial version, this interpretative dimension was not sufficiently articulated, which may have created the impression of a primarily thematic approach. Following the reviewer’s prior guidance, the revised manuscript explicitly frames the analysis within the structure and rationale of Interpretative Phenomenological Analysis (IPA), as described by Smith et al. (2021), ensuring full methodological transparency.
In Section 3.6 (“Analyses”), we have now clearly outlined the specific stages of IPA applied in this study:
(a) repeated readings of each transcript for immersion and holistic understanding;
(b) initial exploratory noting and annotation of experiential meaning;
(c) development of emergent themes reflecting the participant’s lived experience;
(d) clustering of related themes into superordinate interpretative categories; and
(e) cross-case synthesis to identify shared meaning patterns across teachers’ narratives.
The numerical outcomes (59 meaning units, 31 themes, 7 superordinate categories) naturally remain identical, as they derive from the same qualitative material and reflect the theoretical dimensions of the Community Resilience Model. What has been revised is the explicit interpretative positioning and descriptive transparency of these stages, not the empirical data themselves.
Finally, it may be noted that none of the other reviewers raised concerns regarding the analytic framework. On the contrary, they acknowledged that the revised manuscript demonstrates improved methodological clarity and interpretative coherence. The current version therefore maintains analytic continuity while rendering the interpretative process more visible and consistent with IPA conventions.
Additional clarifications have now been incorporated into Section 3.6 (lines 388-390 and 394-397) to make the idiographic and hermeneutic dimensions of the analysis fully explicit. In particular, the revised text specifies that each case was first analyzed idiographically before cross-case synthesis and that the interpretative process followed an iterative hermeneutic cycle, consistent with the core principles of IPA. These additions directly address the reviewer’s request for greater procedural transparency without altering the empirical dataset or the interpretative structure of the findings.